# Zero-shot learning enables instant denoising and super-resolution in optical fluorescence microscopy

Chang Qiao [1,2,3,4,9], Yunmin Zeng [1,9], Quan Meng [5,6,9], Xingye Chen[1,2,3,4,7,9], Haoyu Chen[5,6], Tao Jiang[5,6], Rongfei Wei[5], Jiabao Guo[5,6], Wenfeng Fu[5,6], Huaide Lu[5,6], Di Li [5], Yuwang Wang [8], Hui Qiao [1,2,3,4], Jiamin Wu [1,2,3,4], Dong Li [5,6] ✉ & Qionghai Dai [1,2,3,4] ✉

Computational super-resolution methods, including conventional analytical algorithms and deep learning models, have substantially improved optical microscopy. Among them, supervised deep neural networks have demonstrated outstanding performance, however, demanding abundant high-quality training data, which are laborious and even impractical to acquire due to the high dynamics of living cells. Here, we develop zero-shot deconvolution networks (ZS-DeconvNet) that instantly enhance the resolution of microscope images by more than 1.5-fold over the diffraction limit with 10-fold lower fluorescence than ordinary super-resolution imaging conditions, in an unsupervised manner without the need for either ground truths or additional data acquisition. We demonstrate the versatile applicability of ZS-DeconvNet on multiple imaging modalities, including total internal reflection fluorescence microscopy, three-dimensional wide-field microscopy, confocal microscopy, two-photon microscopy, lattice light-sheet microscopy, and multimodal structured illumination microscopy, which enables multi-color, long-term, super-resolution 2D/3D imaging of subcellular bioprocesses from mitotic single cells to multicellular embryos of mouse and *C. elegans*.

Optical fluorescence microscopy is an essential tool for biological research. The recent developments of super-resolution (SR) techniques provide unprecedented resolvability to visualize the fine dynamic structures of diverse bioprocesses[1]. However, the gain in spatial resolution via any SR method comes with trade-offs in other imaging metrics, e.g., duration or speed, which are equally important for dissecting bioprocesses[1,2]. Recently, computational SR methods have gained considerable attention for their ability to instantly enhance the image resolution in silico[3–12], enabling a significant upgrade of existing fluorescence microscopy systems and extension of their application range.

In general, existing computational SR methods can be classified into two categories: analytical model-based methods such as deconvolution algorithms[4–6], and deep learning-based methods, for

[1]Department of Automation, Tsinghua University, 100084 Beijing, China. [2]Institute for Brain and Cognitive Sciences, Tsinghua University, 100084 Beijing, China. [3]Beijing Key Laboratory of Multi-dimension & Multi-scale Computational Photography, Tsinghua University, 100084 Beijing, China. [4]Beijing Laboratory of Brain and Cognitive Intelligence, Beijing Municipal Education Commission, 100010 Beijing, China. [5]National Laboratory of Biomacromolecules, New Cornerstone Science Laboratory, CAS Center for Excellence in Biomacromolecules, Institute of Biophysics, Chinese Academy of Sciences, 100101 Beijing, China. [6]College of Life Sciences, University of Chinese Academy of Sciences, 100049 Beijing, China. [7]Research Institute for Frontier Science, Beihang University, 100191 Beijing, China. [8]Beijing National Research Center for Information Science and Technology, Tsinghua University, 100084 Beijing, China. [9]These authors contributed equally: Chang Qiao, Yunmin Zeng, Quan Meng, Xingye Chen. ✉e-mail: lidong@ibp.ac.cn; qhdai@tsinghua.edu.cn

example, SR neural networks[7–12]. The former category often employs analytical models prescribing certain assumptions about the specimen and image properties, e.g., sparsity[5] and local symmetry[13,14], to improve the image resolution with multiple tuneable parameters. Parameter tuning is experience-dependent and time-consuming, and the outputs of analytical models greatly depend on the parameter sets[5,13,15,16]. Moreover, in practical experiments, handcrafted models with certain assumptions cannot address the full statistical complexity of microscope imaging, thus lacking robustness and are prone to generate artifacts, especially under low signal-to-noise ratio (SNR) conditions[9]. On the other hand, deep learning-based SR (DLSR) methods have achieved stunning success in learning the end-to-end image transformation relationship according to large amounts of exemplary data without the need for an explicit analytical model[7–12]. Of note, the data-driven inversion scheme via deep learning can approximate not only the pseudoinverse function of the image degradation process but also the stochastic characteristics of the SR solutions. Nevertheless, the training of DLSR models requires acquiring large amounts of paired low-resolution input images and high-quality ground truth (GT) SR images, which are extremely laborious and sometimes even impractical due to the rapid dynamics or the low fluorescence SNR in biology specimens[3,8,17]. In addition, the performance of DLSR methods strongly depends on the quality and quantity of training data[17]. These factors significantly hinder the wide application of DLSR methods in daily imaging experiments despite their compelling SR performance compared to analytical model-based methods[3,17].

Here, we present a zero-shot deconvolution deep neural network (ZS-DeconvNet) framework that is able to train a DLSR network in an unsupervised manner using as few as only one single planar image or volumetric image stack of low-resolution and low-SNR, which results in a zero-shot implementation[18]. As such, compared to state-of-the-art DLSR methods[7–12,19–23], the ZS-DeconvNet can adapt to diverse bioimaging circumstances, where the bioprocesses are too dynamic, too light-sensitive to acquire the ground-truth SR images, or the image acquisition process is affected by unknown and nonideal factors. We characterized that ZS-DeconvNet can improve the resolution by more than 1.5-fold over the diffraction limits with high fidelity and quantifiability, even when trained on a single low SNR input image and without the need for image-specific parameter-tuning[5,13]. We demonstrated that the properly trained ZS-DeconvNet could infer the high-resolution image on millisecond timescale, achieving high throughput long-term SR 2D/3D imaging of multiple organelle interactions, cytoskeletal and organellar dynamics during the light sensitive processes of migration and mitosis, and subcellular structures and dynamics in developing *C. elegans* and mouse embryos. Furthermore, to allow the ZS-DeconvNet to be widely accessible for biology research community, we built up a Fiji plugin toolbox[24] and a tutorial homepage for ZS-DeconvNet methods.

## Results
### Development and characterization of ZS-DeconvNet
The concept of ZS-DeconvNet is based on the optical imaging forward model informed unsupervised inverse problem solver:

$$\text{argmin}_{\boldsymbol{\theta}} \left\| \mathbf{y} - (f_{\boldsymbol{\theta}}(\mathbf{y})^*\text{PSF})_{\downarrow} \right\|_2^2 \quad (1)$$

where **y** denotes the noisy low-resolution image, PSF is the points spread function (PSF), $f_{\boldsymbol{\theta}}$ represents a deep neural network (DNN) with trainable parameters $\boldsymbol{\theta}$, and $(\cdot)_{\downarrow}$ indicates downsampling operation. If the DNN is trained directly via the above objective function, it will undesirably amplify the photon noise contained in the biological images, which will substantially contaminate the real specimen information at low SNR conditions[25] (Supplementary Fig. 1a). To improve the noise robustness of ZS-DeconvNet while maintaining its unsupervised characteristic, we adopted an image recorrupting

scheme[26] that generates two noise-independent recorrupted images from the original image, which are then used as inputs and GTs in the network training (Methods). We theoretically demonstrated the validity of the Gaussian approximation to the mixed Poisson-Gaussian noise model for ordinary sCMOS images and proved the convergency of incorporating the recorrupting scheme into the unsupervised inverse problem solver (Supplementary Note 1). Furthermore, we introduced the Hessian regularization term, which has been demonstrated to be useful for mitigating reconstruction artifacts in microscopy images[27,28], to regulate the network convergence (Supplementary Fig. 1b–e). Taken together, the overall objective function of ZS-DeconvNet can be formulated as:

$$\text{argmin}_{\boldsymbol{\theta}} \frac{1}{N}\sum_{i=1}^{N} \mathcal{L}\Big(\mathbf{y}_i - D^{-1}\mathbf{g}, (f_{\boldsymbol{\theta}}(\mathbf{y}_i + D\mathbf{g})^*\text{PSF})_{\downarrow}\Big) + \lambda\mathcal{R}_{Hessian}\big(f_{\boldsymbol{\theta}}(\mathbf{y}_i + D\mathbf{g})\big) \quad (2)$$

where $N$ is the total number of images to be processed, $D$ is an invertible noise control matrix that can be calculated according to the signal and noise levels (Methods), and **g** is a random noise map that is sampled from a standard normal distribution. We refer to the first part of the objective function as the degradation term, which accounts for the inference fidelity, and the second part as the regularization term, rationalizing the SR outputs.

After defining the objective function, we adopted a dual-stage DNN architecture composed of two sequentially connected U-Nets[29] as a simple but effective backbone for ZS-DeconvNet (Fig. 1a, b and Supplementary Fig. 2a). The first stage serves as a denoiser to generate noise-free images according to the denoising loss (Methods), and the second stage enhances the image resolution according to the unsupervised deconvolution loss described above. We empirically found that the dual-stage architecture and the physical model-regulated loss function stabilize the training procedures and endow interpretability for the overall network model.

To characterize and evaluate ZS-DeconvNet, we first simulated the microscopy images of punctate and tubular structures contaminated by Gaussian-Poisson noise at escalating signal levels from 5 to 25 average photon counts, which allowed us to systematically test how the recorrupting hyperparameter settings at different imaging conditions influence the final outputs (Supplementary Note 2). We found that the optimal hyperparameters are theoretically independent of the image contents and signal levels (Supplementary Figs. 3–5), thus enabling a robust application of ZS-DeconvNet onto various biological specimens and imaging configurations (Supplementary Note 4). Next, we compared the performance of the ZS-DeconvNet models trained with the data augmented by recorrupting a single noisy image with analytical deconvolution algorithms or the models trained with numbers of simulated or independently acquired images. To do so, we employed the total internal reflective fluorescence (TIRF) illumination mode of our home-built multimodal structured illumination microscopy (Multi-SIM)[8,30] to acquire ~20 sets of diffraction-limited TIRF images at low- and high-SNR for each subcellular structure of lysosomes (Lyso) and microtubules (MTs), of which the low-SNR images were used for training and testing, while their high-SNR counterparts served as reference (Methods). We found that the peak signal-to-noise ratio (PSNR) and resolution of ZS-DeconvNet images were substantially better than those generated by analytical algorithms, such as the classic Richardson-Lucy (RL) and the latest developed sparse deconvolution[5] (Fig. 1c–e) and the throughput rate of a well-trained ZS-DeconvNet is >100-fold higher than that of the sparse deconvolution algorithm (Fig. 1f). In particular, even if the ZS-DeconvNet was trained with the augmented data from a single input image, the perceptual quality and quantified metrics of its output images were comparable with the images from the model trained with larger amounts of data (Supplementary Fig. 6). Furthermore, we validated the resolution improvement, quantifiability, and the generalization capability of ZS-DeconvNet (Supplementary Figs. 7–10), and

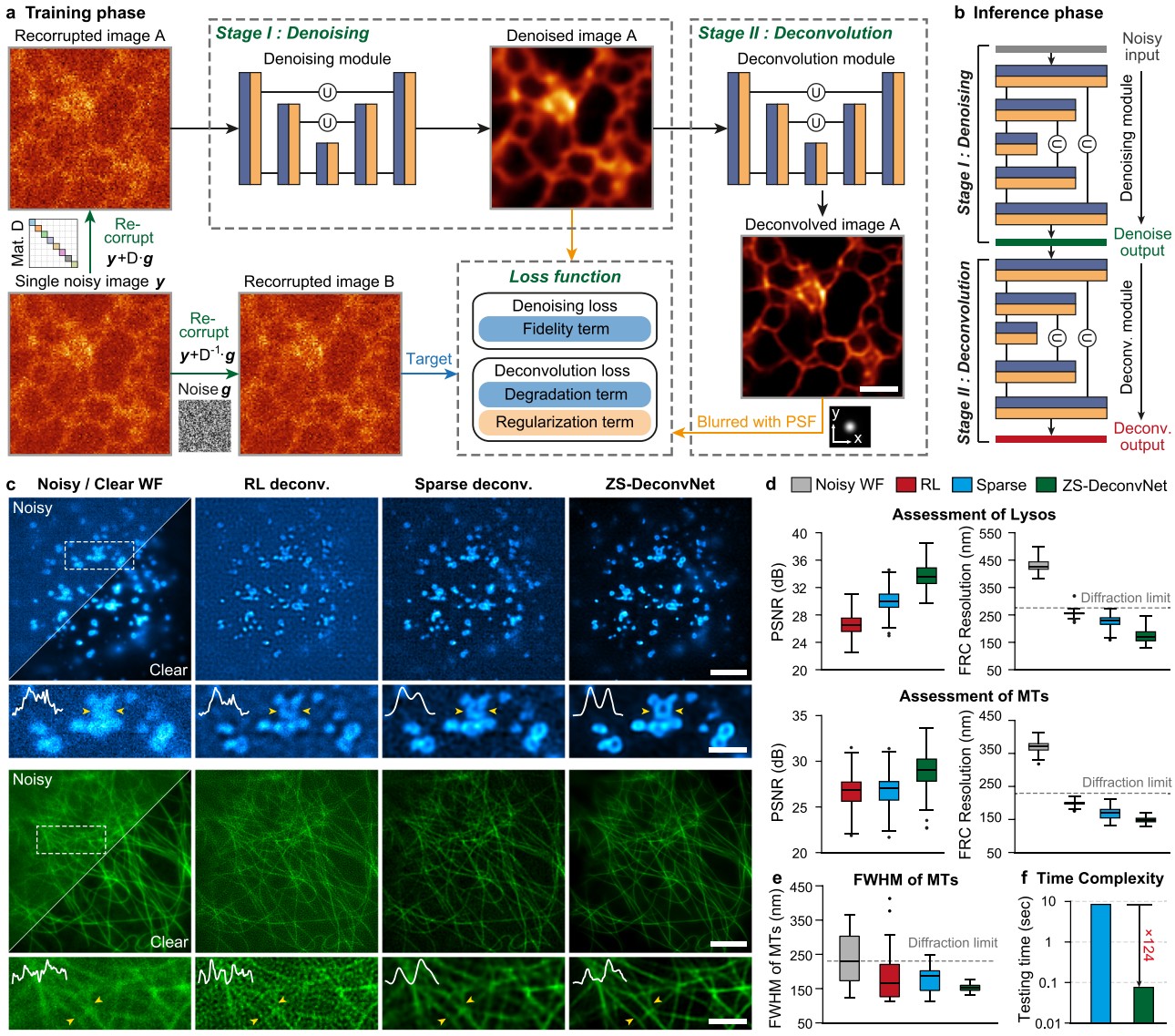

**Fig. 1 | Zero-shot deconvolution networks. a** The dual-stage architecture of ZS-DeconvNet and the schematic of its training phase. **b** The schematic of the inference phase of ZS-DeconvNet. **c** Representative SR images of Lyso and MTs reconstructed by RL deconvolution (second column), sparse deconvolution (third column) and ZS-DeconvNet (fourth column). The clear WF images are displayed for reference. **d** Statistical comparisons of RL deconvolution, sparse deconvolution and ZS-DeconvNet in terms of PSNR and resolution ($n = 100$ regions of interest). **e** Full width at half maximum (FWHM) comparisons of clear WF images and processed images via RL deconvolution, sparse deconvolution and ZS-DeconvNet

($n = 30$ microtubules). The theoretical diffraction limit is labeled with the gray dashed line for reference. **f** Testing time comparison between GPU-based sparse deconvolution and ZS-DeconvNet (average from 25 testing images of 1024 × 1024 pixels). Center line, medians; limits, 75% and 25%; whiskers, the larger value between the largest data point and the 75th percentiles plus 1.5× the interquartile range (IQR), and the smaller value between the smallest data point and the 25th percentiles minus 1.5× the IQR; outliers, data points larger than the upper whisker or smaller than the lower whisker. Source data are provided as a Source Data file. Scale bar, 1.5 μm (**a**), 5 μm (**c**), 2 μm (zoom-in regions in (**c**)).

compared it with the supervised DFCAN model[8] (Supplementary Fig. 11) on synthetic and experimental data. These characterizations demonstrate that ZS-DeconvNet is able to generate high-quality DLSR images of 1.5-fold resolution improvement relative to the diffraction limit while using the least training data, which holds great potential to upgrade the imaging performance of diverse microscope systems, and extend their applicability into a wide variety of bioprocesses that are challenging for conventional methods.

## Long-term observation of bioprocesses sensitive to phototoxicity

Cell adhesion and migration are essential in morphogenetic processes and contribute to many diseases[31]. Visualizing cytoskeletal dynamics at high resolution during the adhesion/migration process is critical for

elucidating the underlying mechanism. However, due to severe photosensitivity, the whole processes of cell adhesion and migration are typically recorded at low framerates, i.e., several seconds per frame, and low light intensities[9,32]. Under these imaging conditions, either RL deconvolution or temporal continuity-based self-supervised learning[33] (Methods) fails to recover and sharpen the intricate structure of F-actin and myosin-II (Fig. 2a, Supplementary Fig. 12, and Supplementary Video 1). In contrast, the ZS-DeconvNet model effectively improves both the SNR and resolution of the two-color time-lapse recordings of cell

spreading processes after dropping a cell coexpressing mEmerald-Lifeact and mCherry-myosin- IIA onto a coverslip (Fig. 2b and Supplementary Video 2). Intriguingly, we observed that in certain substances cells crawled around the contact site to explore the

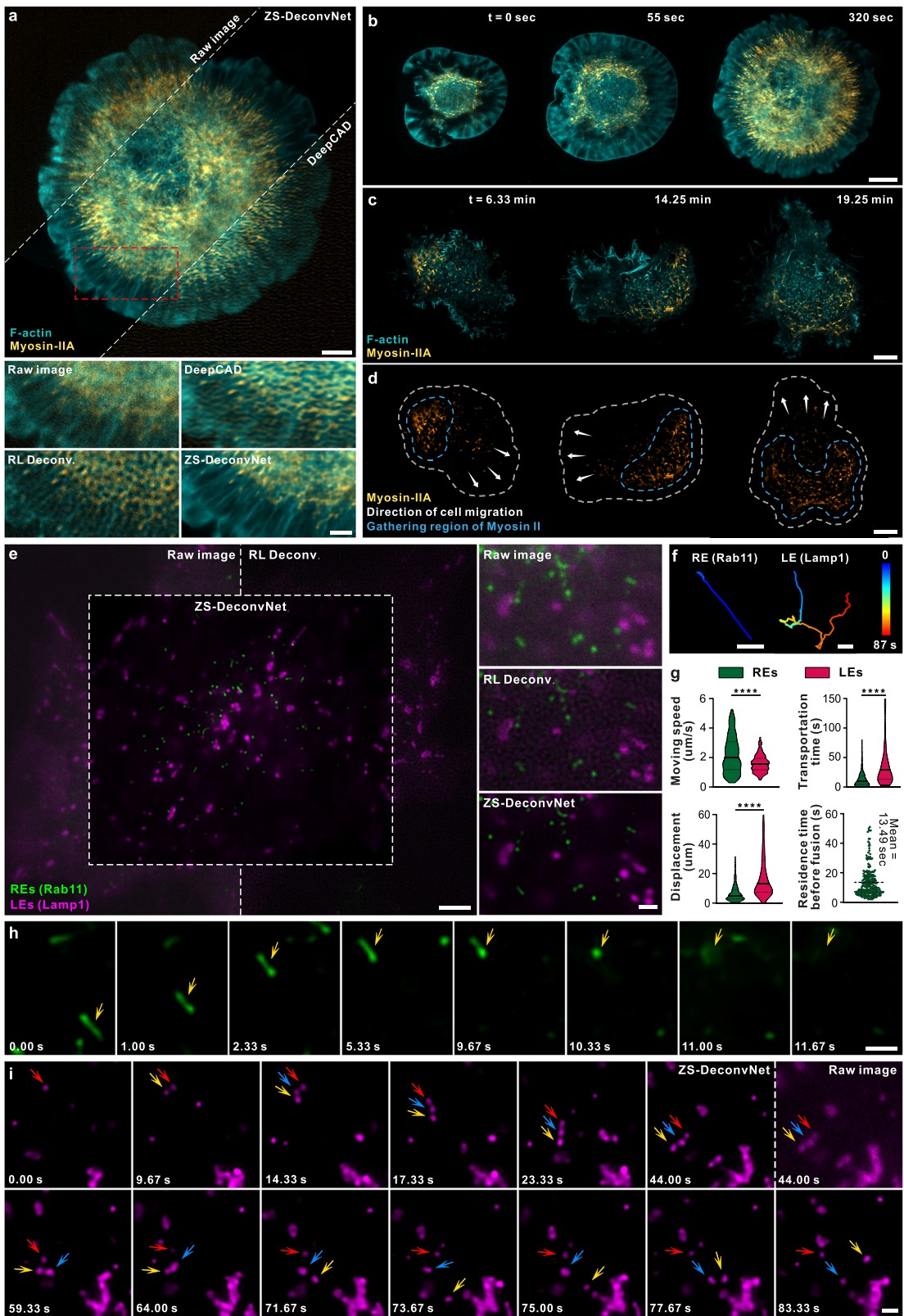

neighborhood before spreading and adhering (Fig. 2c and Supplementary Video 3). The cell crawling was preceded by the polarized accumulation of myosin-II at the cell rear, leading to cell migration in the opposite direction driven by posterior myosin-II contractility. Moreover, the migration direction could be swiftly changed in response to the dynamic redistribution of myosin-II within the cell (Fig. 2d). These results demonstrate that the kinetics of cell adhesion

and migration can be faithfully recorded by ZS-DeconvNet-assisted imaging without perturbing this lengthy and vulnerable process.

**Visualizing the rapid dynamics of the endolysosomal system**
The endolysosomal system includes diverse types of vesicles that function in a highly dynamic, yet well-organized manner. Although live-cell fluorescence imaging has remarkably improved our

**Fig. 2 | Long-term SR imaging of rapid and photo-sensitive bioprocesses via ZS-DeconvNet. a** Representative SR images reconstructed by ZS-DeconvNet of F-actin cytoskeleton and myosin-II in a COS-7 cell co-expressing mEmerald-lifeact and mCherry-myosin-IIA. Comparisons of raw noisy TIRF image and images processed by RL deconvolution, DeepCAD-based deconvolution and ZS-DeconvNet are displayed. **b** Two-color time-lapse SR images enhanced via ZS-DeconvNet showing the coordinated dynamics of F-actin (cyan) and myosin-II (yellow) over the whole spreading process after placing a COS-7 cell onto a coverslip (Supplementary Video 2). **c, d** Two-color time-lapse SR images enhanced via ZS-DeconvNet of F-actin and myosin-II in a crawling COS-7 cell showing that myosin-II preferentially concentrates to the rear of the cell (outlined by yellow dashed lines in **d**), opposite to crawling direction (indicated by the white arrows in **d**) (Supplementary Video 3). **e** Representative SR image generated via ZS-DeconvNet of recycling endosomes (REs, green) and late endosomes (LEs, magenta) in a gene-edited SUM-159 cell endogenously expressing EGFP-Rab11 and mCherry-Lamp1 (Supplementary Video 4). **f** Typical trajectories of RE (top) and LE (bottom) movements showing the rapid directional motility of RE, and the bidirectional nature of LE. **g** Comparisons of the speed, displacement, and traveling time between Lyso/LEs and REs, and quantification of the residence time of REs near their exocytosis sites before fusing with plasma membrane ($n = 505$ tracks for REs and $n = 230$ tracks for LEs). A small number of data points exceeding transportation time of 150 s or displacement of 60 μm were not displayed for better presentation of the distributions. Center line, medians; limits, 75% and 25%. Statistical significance was determined using unpaired Mann-Whitney test (p = $1.38 \times 10^{-7}$, $5.65 \times 10^{-35}$, and $6.26 \times 10^{-40}$ for tests of the moving speed, transportation time, and displacement, respectively). ****$p < 0.0001$. Source data are provided as a Source Data file. **h** Time-lapse images illustrate the directional movement of a RE in rod-like shape, and the subsequent fusion with plasma membrane. **i** Time-lapse images illustrate three LEs tether each other and co-migrate for certain distance before splitting into individual LEs. Scale bar, 5 μm (**a, c, d**), 2 μm (zoom-in regions in **a**), 8 μm (**b**), 3 μm (**e**), 0.5 μm (zoom-in region in **e**), 1 μm (**g, f, i**).

understanding of the endolysosomal system, most studies had to overexpress the proteins of interest to record their rapid dynamics[30], which often resulted in artifact morphologies or behaviors. With ZS-DeconvNet, we were able to image the knock-in SUM-159 cell line endogenously expressing EGFP-Rab11 and mCherry-Lamp1 for 1,500 frames at ~150 nm resolution and 3 frames per second in two colors (Fig. 2e and Supplementary Video 4), thereby allowing us to visualize and track the rapid motion of recycling-endosomes (REs) and lysosomes or late endosomes (LEs) on a substantially finer spatiotemporal scale and longer observation window than previously achieved[34]. As exemplified in Fig. 2f–h, we found that the majority of REs ($n = 505$ tracks) experienced a directional movement, with a total displacement of $6.7 \pm 5.4$ μm at a high speed of $2.2 \pm 1.2$ μm/s (instantaneous speed exceeding 5.3 μm/s), with a rare intermediate pause, then stopped at specific sites for a period of $13.5 \pm 10.3$ sec before fusing with the plasma membrane. This observation suggests that REs might be efficiently transported over long ranges to regions near the plasma membrane to facilitate subsequent exocytosis. Unexpectedly, ZS-DeconvNet captured multiple fission events of the Rab11-positive REs, in which both separated REs underwent exocytosis sequentially (Supplementary Fig. 13a) or one RE moved away (Supplementary Fig. 13b). This observation indicates that the highly specialized Rab11-positive REs may be subject to further cargo sorting right before exocytosis.

In contrast, the movements of LEs were typically discontinuous and proceeded in a bidirectional stop-and-go manner at a relatively slow speed of $1.6 \pm 0.6$ μm/s ($n = 230$ tracks) (Fig. 2f, g, i). Although the transportation of LEs seemed inefficient, the LEs often persisted for a long period of 91.8 s with a total displacement as long as 23.6 μm (averaged from $n = 230$ tracks) (Fig. 2h). Interestingly, we noticed that two or more LEs sometimes tended to tether each other in a kiss-and-stay fashion and migrate for a certain distance before splitting into individual LEs again (Fig. 2i and Supplementary Fig. 13c), which might facilitate the directional movement of LEs without sufficient motor-protein-adaptors for long-range transportation. These complex dynamics of LEs suggest that their positioning and mobility are delicately regulated by multiple factors, such as MT-based motors and membrane contacts.

### 3D ZS-DeconvNet for lattice light-sheet microscopy

Volumetric live-cell imaging conveys more biological information than 2D observations; however, it is subject to much severer phototoxicity, photobleaching and out-of-focus fluorescence contamination. To extend the superior capability of ZS-DeconvNet to volumetric SR imaging, we upgraded the backbone of the dual-stage network architecture into a 3D RCAN, which has been demonstrated to be suitable for volumetric image restoration[9,35] (Fig. 3a, b and Supplementary Fig. 2b). Next, we integrated our previously proposed spatially

interleaved self-supervised learning scheme[9] with the physical model-informed self-supervised inverse problem solver to construct the 3D ZS-DeconvNet. The 3D ZS-DeconvNet with spatially interleaved self-supervised scheme follows a simpler data augmentation procedure (Methods), while achieving comparative or even better performance than the recorruption-based strategy (Supplementary Fig. 14).

We systematically assessed the 3D ZS-DeconvNet model with datasets of three different biological specimens acquired via our home-built lattice light-sheet structured illumination microscopy[36] (LLS-SIM), in which the diffraction-limited data acquired by the lattice light-sheet microscopy (LLSM) mode was used for training while the SR counterparts acquired by the LLS-SIM mode served as references (Methods). We found that 3D ZS-DeconvNet successfully reconstructed the elaborate filaments of F-actin, the hollow structure of the mitochondrial (Mito) outer membrane, and the intricate networks of the endoplasmic reticulum (ER) with high fidelity and resolution comparable to LLS-SIM images acquired under high-SNR conditions (Fig. 3c). The quantifications of PSNR and resolution illustrate that the 3D ZS-DeconvNet model substantially outperforms conventional analytical model-based approaches in diverse biological specimens (Fig. 3d). We demonstrate that by training with the noisy image stacks themselves, the dual-stage 3D ZS-DeconvNet not only generated denoised results comparable to state-of-the-art self-supervised denoising techniques[37,38] (Supplementary Fig. 15), but also provided super-resolved image stacks with significant resolution improvement by over 1.5-fold both laterally (Supplementary Fig. 16) and axially (Supplementary Fig. 17). Moreover, by sequentially incorporating self-learning-based axial resolution-enhancement methods[39,40], the axial resolution can be improved further (Supplementary Fig. 17g–i).

### Long-term volumetric super-resolution imaging enabled by 3D ZS-DeconvNet

Volumetric observation of cell division at high spatiotemporal resolution is of vital importance for exploring mitosis-related biological mechanisms, such as the mechanism that allocates the numerous distinct organelles in the cytoplasm into each daughter cell[41,42]. Due to the extreme light sensitivity and vulnerability of mitotic cells, previous volumetric SR imaging of this process has relied on the low-light LLS-SIM system and supervised learning-based SR reconstruction[9]. However, collecting high-quality training data is extremely laborious and sometimes impractical because the morphology and distribution of organelles usually undergo dramatic changes during mitosis[41]. Here, we demonstrate that the self-supervised 3D ZS-DeconvNet model can be generally applied to superresolve the fine subcellular structures of the ER, Mito, and chromosomes from noisy LLSM volumes without the need for additional training data, thus enabling fast and long-term volumetric SR observation of multiple organelles for 1,000 timepoints at 10 sec

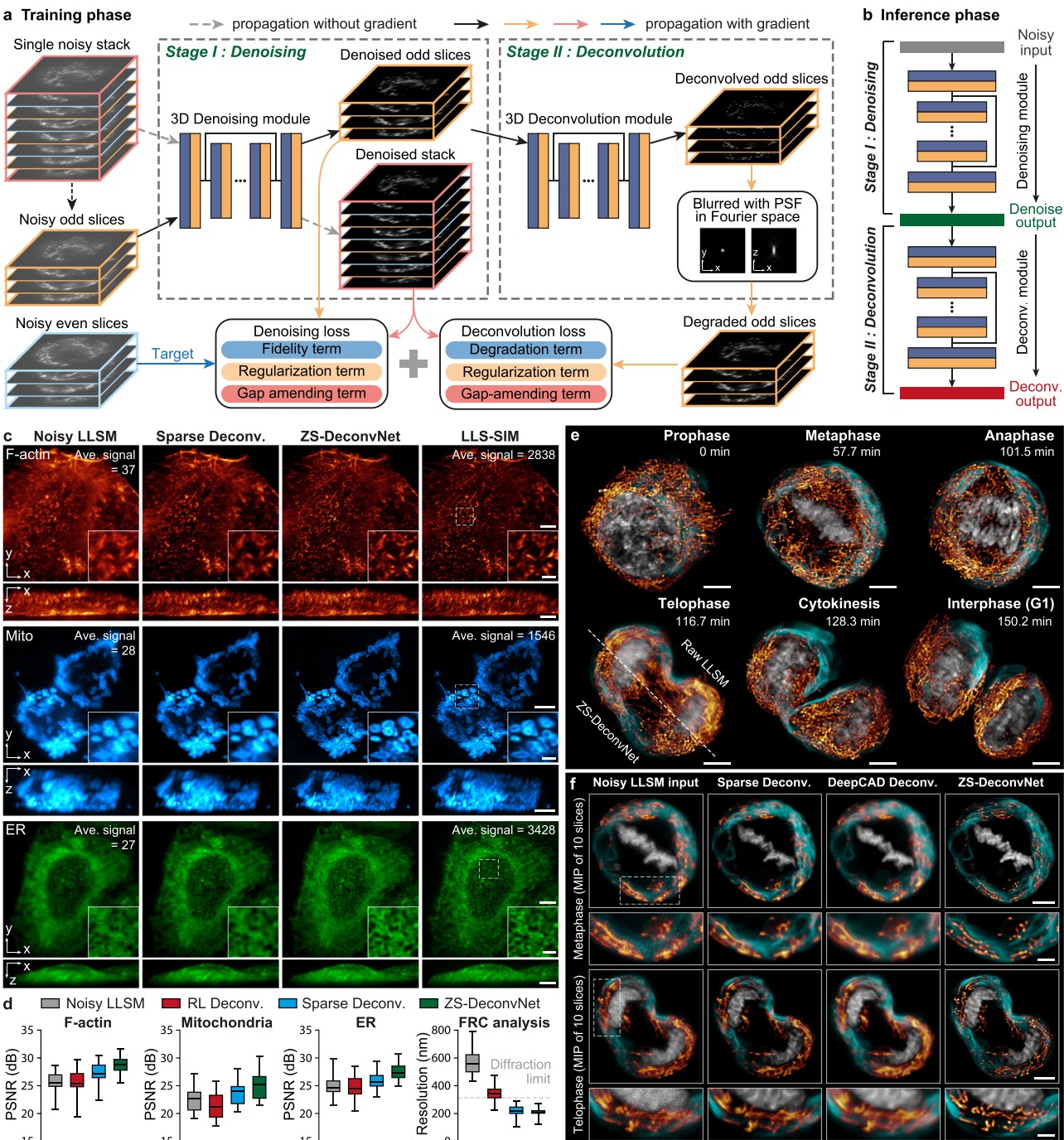

**Fig. 3 | Characterizations and demonstrations of 3D ZS-DeconvNet. a** The network architecture of 3D ZS-DeconvNet and the schematic of its training phase. **b** The schematic of the inference phase of 3D ZS-DeconvNet. **c** Representative maximum intensity projection (MIP) SR images of F-actin, Mito outer membrane, and ER reconstructed by sparse deconvolution (second column), 3D ZS-DeconvNet (third column), and LLS-SIM (fourth column). Average sCMOS counts of the highest 1% pixels for raw images before processing are labeled on the top right corner. **d** Statistical comparisons of RL deconvolution, sparse deconvolution and ZS-DeconvNet in terms of PSNR and resolution on different specimens ($n = 40$ regions of interest). The resolution was measured by Fourier ring correlation analysis[74] with F-actin image stacks. Center line, medians; limits, 75% and 25%; whiskers, maximum and minimum. Source data are provided as a Source Data file. **e** Time-lapse three-color 3D rendering images reconstructed via 3D ZS-DeconvNet of ER, H2B, and Mito, showing their transformations in morphology and distribution as well as interaction dynamics during mitosis (Supplementary Video 5). **f** Representative three-color images obtained with conventional LLSM (first column), sparse deconvolution (second column), DeepCAD based deconvolution (third column) (Methods), and 3D ZS-DeconvNet (fourth column). The comparisons are performed on two typical timepoints of the time-lapse data shown in (**e**). Scale bar, 5 μm (**c**, **e**, **f**), 1.5 μm (zoom-in regions of **c**), 2 μm (zoom-in regions of **f**).

intervals in a mitotic HeLa cell (Fig. 3e and Supplementary Video 5). Moreover, the unsupervised property of ZS-DeconvNet allows us to integrate a test-time adaptation learning strategy[43] to fully exploit the structural content in each noisy volume, which yielded the best 3D SR performance (Methods). In contrast, the conventional prior-dependent deconvolution algorithm[5] and temporally interleaved self-supervised learning[9,33,44] method both failed to restore the high-frequency details of the specimens because of the low SNR condition and weak temporal consistency between adjacent timepoints (Fig. 3f and Methods). Furthermore, according to the low invasiveness provided by 3D ZS-

DeconvNet, a group of mitotic HeLa cells labeled with H2B-mCherry and HeLa-mEmerald-SC35 were imaged in a large field of view (FOV) of 100×50×25 μm³ for more than 300 timepoints, thereby recording the entire disassembly and reassembly processes of nuclear speckles at a high spatiotemporal resolution (Supplementary Fig. 18 and Supplementary Video 6). In brief, 3D ZS-DeconvNet allows biologists to easily explore various light-sensitive bioprocesses with low invasiveness at substantially higher spatiotemporal resolution without the need for any additional datasets or optical setup modifications.

## ZS-DeconvNet for confocal and wide-field microscopy

The ZS-DeconvNet relies on the randomness of noises and the low-pass filter characteristic of optical microscopes, which are common for various types of microscopy modalities. On this basis, we expect that ZS-DeconvNet can be generally applied to all microscopy, e.g., the most commonly used confocal microscopy and wide-field (WF) microscopy. To investigate the performance of 3D ZS-DeconvNet on confocal data, we employed our home-built confocal microscope to acquire a four-color volume of the mouse early embryo immunostained for the microtubule, chromosomes, actin, and apical domain (Methods), which play key roles in the first cell fate decision and are critical for embryo development[45–47]. We then trained 3D ZS-DeconvNet models on this single noisy volume and processed the original data with the trained models. As shown in Figs. 4a, b, 3D ZS-DeconvNet significantly enhances the SNR, contrast, and resolution of

the confocal data volume and resolves the fine structures of microtubule bridges and actin rings (Fig. 4c, d, Supplementary Fig. 19, and Supplementary Video 7). These results indicate that ZS-DeconvNet enables a higher spatial resolution at a lower photon budget for confocal microscopy in imaging specimens on large scale, e.g., mouse early embryos, which is critical to research on cell polarity[47], intracellular transport and blastocyst formation[46].

We next imaged Caenorhabditis elegans embryos with apical junctions, cell membranes and lysosomes marked using the 3D WF mode of our Multi-SIM system (Methods). To ensure that *C. elegans* embryo development was not disturbed, we acquired raw image stacks at relatively low light excitation in intervals of 30 s for more than 200 timepoints. However, under such conditions, the WF images are heavily contaminated by both out-of-focus background and noise (Fig. 4e, f). Even in this challenging situation, 3D ZS-DeconvNet images presented considerable suppression upon noise and background while enhancing the spatial resolution of the subcellular details (Fig. 4e, g and Supplementary Video 8), thus allowing us to investigate the elaborate process of embryonic development, e.g., hypodermal cell fusion[48] (Fig. 4h), even via a simple WF microscope.

## ZS denoising and resolution enhancement in multimodal SIM images

Among the various forms of SR microscopy, structured illumination microscopy (SIM) is often recognized as a balanced option for SR

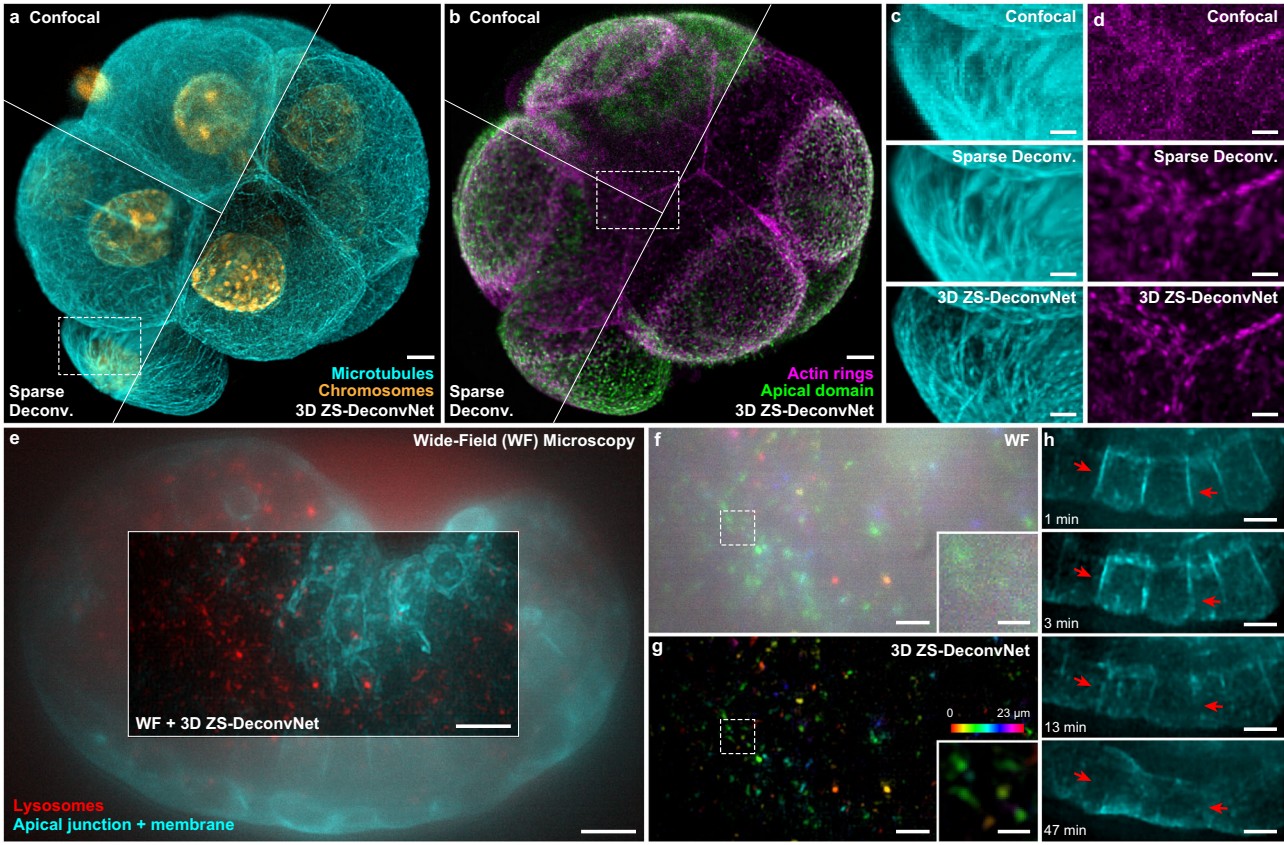

**Fig. 4 | Generalization of ZS-DeconvNet to multiple imaging modalities.**
**a**, **b** Representative confocal (top left), sparse deconvolution (bottom left), and 3D ZS-DeconvNet enhanced (right) images of an early mouse embryo immunostained for microtubule (cyan), chromosomes (orange), actin rings (magenta), and apical domain (green). **c**, **d** Magnified regions of microtubule bridges (c) and actin rings (d) labeled with white dashed boxes in (**a**) and (**b**) acquired via confocal microscopy, sparse deconvolution, and 3D ZS-DeconvNet. **e** Representative WF (center region) and 3D ZS-DeconvNet enhanced (surrounding region) images of a *C.*

*elegans* embryo with apical junction, cell membrane (cyan) and lysosomes (red) labeled. **f**, **g** Lysosome channel of the central region in (**e**) color-coded for distance from the substrate. Both WF (**f**) and 3D ZS-DeconvNet processed images (**g**) are shown for comparison. **h** Time-lapse 3D ZS-DeconvNet enhanced images showing the process of hypodermal cell fusion (red arrows) during the development of a *C. elegans* embryo. Scale bar, 5 μm (**a**, **b**, **e**), 2 μm (**c**, **d**), 3 μm (**g**, **h**), 1 μm (zoom-in region of **g**). Gamma value, 0.7 for cytomembrane and lysosomes in the *C. elegans* embryo.

live-cell imaging because it needs less than ten raw modulated images to provide a twofold improvement in spatial resolution[1,2]. Nevertheless, conventional SIM has two critical limitations: first, further resolution enhancement requires considerably more raw data, i.e., at least 25 raw images are needed for nonlinear SIM to obtain a sub-80 nm resolution[49,50]; second, the postreconstruction of SIM images generally requires raw images with a high SNR to eliminate noise-induced reconstructed artifacts, thus impairing fast, low-light, and long-term live-cell imaging[51]. Recent studies have explored supervised learning approaches by either denoising SIM images[9,52] or reconstructing SR SIM images directly from noisy raw images[8,22] to achieve low-light SIM reconstruction; however, these methods require abundant training data and do not further enhance the resolution. In light of the superb denoising and SR capability of ZS-DeconvNet, we integrated the zero-shot learning scheme with the conventional SIM reconstruction algorithm, and theoretically proved that ZS-DeconvNet is suitable for processing the SR-SIM images (Supplementary Note 1). We designed the ZS-DeconvNet enhanced SIM (ZS-DeconvNet-SIM) model to simultaneously denoise

and sharpen SR SIM images in an unsupervised manner (Fig. 5a, Supplementary Fig. 20a, and Methods). Resorting to the remarkable improvement in both SNR and resolution provided by ZS-DeconvNet-SIM (Supplementary Figs. 21, 22), the hollow structure of clathrin-coated pits (CCPs) in a SUM-159 cell and the densely interlaced cytoskeletons in a COS-7 cell, which are indistinguishable in WF and conventional SIM images, were clearly resolved (Fig. 5b, c). Moreover, we demonstrated that the ZS-DeconvNet-SIM can be applied in 3D-SIM modality to simultaneously denoise and sharpen the 3D-SIM images in both lateral and axial axes (Methods, Supplementary Fig. 23).

Furthermore, we integrated 3D ZS-DeconvNet with LLS-SIM to develop the 3D ZS-DeconvNet-SIM modality (Supplementary Fig. 20b). By incorporating the anisotropic PSF of conventional LLS-SIM[36] into the training process, 3D ZS-DeconvNet LLS-SIM not only prominently enhanced the contrast and resolution in all three dimensions but also provided an approximately isotropic lateral resolution of ~150 nm (Fig. 5d, e, and Supplementary Fig. 22). These successful applications of ZS-DeconvNet to multimodal SIM systems demonstrate its

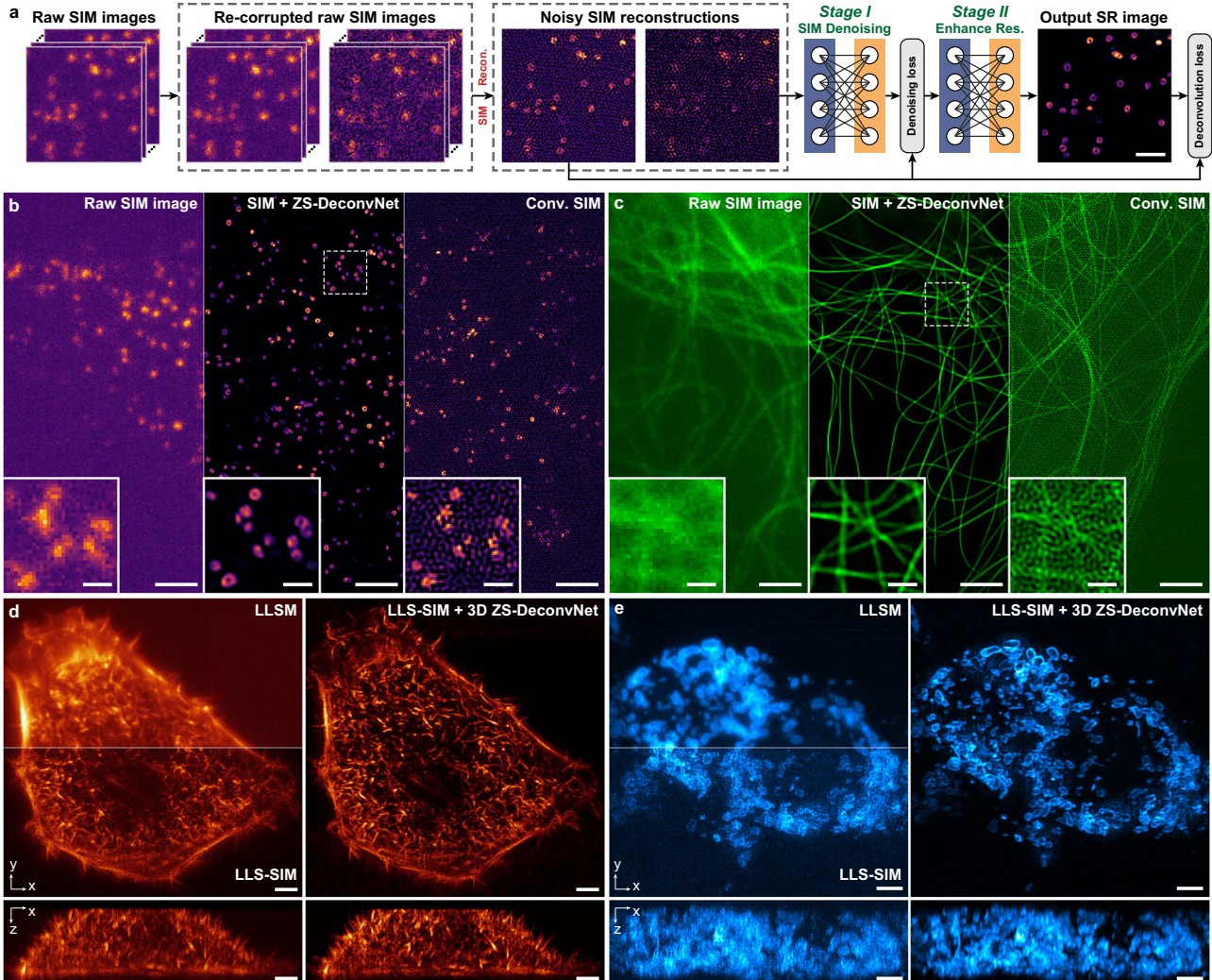

**Fig. 5 | Zero-shot denoising and resolution enhancement in multimodal SIM data. a** Schematic of the training procedure of ZS-DeconvNet for SIM. **b** Progression of SNR and resolution improvement across the CCPs in a SUM-159 cell, from raw SIM images (left), conventional SIM image (right), and ZS-DeconvNet enhanced SIM image (middle). **c** Progression of SNR and resolution improvement across the microtubules in a COS-7 cell, from raw SIM images (left), conventional SIM image (right), and ZS-DeconvNet enhanced SIM image (middle).

**d** Representative maximum intensity projection (MIP) images of F-actin in a HeLa cell obtained via LLSM, LLS-SIM, and LLS-SIM enhanced by 3D ZS-DeconvNet across three dimensions. **e**, Representative MIP images of mitochondrial outer membrane labeled with TOMM20 in a 293 T cell obtained via LLSM, LLS-SIM, and LLS-SIM enhanced by 3D ZS-DeconvNet across three dimensions. Scale bar, 1 μm (**a**), 2 μm (**b**, **c**), 0.5 μm (zoom-in regions in **b**, **c**), 3 μm (**d**, **e**).

capability to further extend the spatiotemporal resolution bandwidth of existing SR techniques.

## Discussion

The ultimate goal of live imaging is to collect the most spatiotemporal information about bioprocesses with the least invasiveness to biological specimens. However, the mutual restrictions between imaging speed, duration, resolution, and SNR in fluorescence microscopy together result in the spatiotemporal bandwidth limitation[53], which limits the synergistic improvement in all these aspects. For instance, to obtain higher spatial resolution, conventional SR techniques have to rely on repetitive acquisitions or additional excitation[1], which aggravates phototoxicity and photobleaching, impeding fast, long-term observations of bioprocesses. To address the spatiotemporal bandwidth limitations in microscopy, we made an in-depth analysis of noise propagation in the optical imaging model and SIM reconstruction (Supplementary Note 1), proved the convergence of the recorruption-integrated self-supervised loss function in both ordinary and SIM scenarios based on the linearity of PSF convolution, and proposed the versatile ZS-DeconvNet framework, which can be incorporated with various optical fluorescence microscope to instantly enhance image SNR and resolution without compromising other imaging properties. We emphasize that the application of ZS-DeconvNet is robust to the hyperparameters in image recorruption process (Supplementary Fig. 24) and that ZS-DeconvNet can be well trained with only one slice or stack of raw images (Supplementary Figs. 6, 16) without using assumptions of structural sparsity[5] and temporal continuity[28,33,44]. The qualitative and quantitative evaluations on both simulated and experimental data show that our methods substantially enhance the image quality and resolution by more than 1.5-fold with high fidelity and quantifiability even under low-light conditions, thereby permitting fast, long-term, super-resolution observations of multiple subcellular dynamics.

The proposed ZS-DeconvNet method has wide functionality for various types of imaging modalities, from scanning-based microscopy, e.g., confocal microscopy and two-photon microscopy (Supplementary Fig. 25), to wide-field detection-based microscopy, e.g., TIRF, 3D WF microscopy, LLSM, and multimodal SIM. We demonstrate its capabilities with more than 10 distinct fixed- or live- specimens imaged via six different microscopy setups, including planar and volumetric imaging of multiple organelles in single cells, observations of subcellular dynamics and interactions during cell mitosis, and multi-color 3D imaging of early mouse embryos and *C. elegans* embryos. To make our methods more accessible and convenient to use, we integrated ZS-DeconvNet and 3D ZS-DeconvNet into a user-friendly Fiji plugin (Supplementary Figs. 26, 27, Supplementary Notes 3, 4, and Supplementary Video 9), allowing users even without deep learning experience to easily train their own ZS-DeconvNet models and enhance microscopy images opened in Fiji in some mouse clicks. The functionality and convenience of ZS-DeconvNet demonstrate its great potential in upgrading the performance of existing optical microscopy.

Despite its general robustness and applicability, users of ZS-DeconvNet should carefully consider the potential appearance of hallucinations and its limitations. First, ZS-DeconvNet may mistake extremely low fluorescence signals as photon noise, thereby weakening them in the output images (Supplementary Fig. 28a). This kind of errors could be detected to some extent via image quality-check tools such as SQUIRREL[54]. Second, if a well-trained ZS-DeconvNet model is applied to processing images significantly different from the training data, e.g., acquired with a different imaging modality, there might be noticeable performance degradation and higher risk of hallucination generation (Supplementary Fig. 28b). Third, ZS-DeconvNet models should be trained using matched PSFs to the dataset, otherwise improper training with mismatched PSFs might result in

inconspicuous resolution improvement or ringing artifacts (Supplementary Fig. 28c). Finally, we do not expect the unsupervised ZS-DeconvNet to generate SR images as good as supervised DLSR models trained with high-quality dataset (Supplementary Fig. 11). However, in imaging experiments when such dataset is not available, ZS-DeconvNet will be a powerful and convenient tool to resolve biological details as fine as possible.

Several improvements and extensions of ZS-DeconvNet can be envisioned. First, we adopted commonly used U-Net and RCAN as the backbone models in our experiments for conceptual demonstration. Combining the ZS-DeconvNet framework with more advanced network architectures such as Richardson-Lucy network, which incorporates the image formation process to accelerate SR information extraction[12], may further improve the SR capability with higher computation efficiency. Second, although we only presented the applications of ZS-DeconvNet on SIM, it can be reasonably speculated that other optics-based SR techniques, such as photoactivated localization microscopy[55], stimulated emission depletion microscopy[56], and image scanning microscopy[57], can be improved by integrating ZS-DeconvNet into their image processing pipelines. Third, due to the lack of generalization, users need to train a specialized model for each type of specimen to achieve the best performance. Incorporating domain adaptation[43] or domain generalization[58] techniques with our methods may effectively alleviate the burden of applying trained models into unseen domains. Finally, we used a spatially invariant PSF for the well-calibrated imaging systems in this work. With spatially varying PSF, the functionality of ZS-DeconvNet can be further extended to various image processing tasks, such as phase space light-field reconstruction and digital adaptive optics[59].

## Methods
### Multi-SIM system
The Multi-SIM system was built based on an invented fluorescence microscope (Ti2E, Nikon). Three laser beams of 488 nm (Genesis-MX-SLM, Coherent), 560 nm (2RU-VFL-P-500-560, MPB Communications), and 640 nm (LBX-640-500, Oxxius), were combined collinearly, and then passed through an acousto-optic tunable filter (AOTF, AOTFnC-400.650, AA Quanta Tech), which serves to select the desired laser wavelength and control its power and exposure time. Afterwards the selected laser light was expanded and sent into an illumination modulator, which is composed of a ferroelectric spatial light modulator (SLM, QXGA-3DM, Forth Dimension Display), a polarization beam splitter, and an achromatic half-wave plate. Different illumination modes were generated by adjusting the patterns displayed on the SLM, e.g., grating patterns of 3-phase × 3-orientation at 1.41 NA for TIRF-SIM or 1.35 NA for GI-SIM. Next, the modulated light was passed through a polarization rotator consisting of a liquid crystal cell (Meadowlark, LRC-200) and a quarter-wave plate, which rotated the linear polarization to maintain the necessary s-polarization, thus maximizing the pattern contrast for all pattern orientations. The diffraction orders, except for ±1 orders for TIRF/GI-SIM, were filtered out by a spatial mask, and then relayed onto the back focal plane of the objectives (1.49 NA, Nikon). The raw SIM images excited by different illumination patterns were sequentially collected by the same objective, then separated by a dichroic beam splitter (Chroma, ZT405/488/560/647tpc), finally captured with an sCMOS camera (Hamamatsu, Orca Flash 4.0 v3). For live imaging, cells were held in a stage top incubator equipped on the microscopy (OkO Lab, H301) to maintain condition at 37°C with 5% $CO_2$. The Multi-SIM system worked in the TIRF mode and 3D WF mode in the experiments shown in Figs. 1, 2 and 4e–h by adjusting the patterns displayed on the SLM to generate uniform TIRF or WF illumination, and worked in the TIRF/GI-SIM mode and 3D-SIM mode in Fig. 5b, c and Supplementary Figs 21-24. Besides TIRF, 3D WF, TIRF/GI-SIM, and 3D-SIM modes used in this work, the Multi-SIM system integrated diverse SIM modalities including nonlinear-SIM and

stacked slice-SIM into a single setup, which has been commercially available from NanoInsights Inc. (nanoinsights-tech.com).

## LLS-SIM system

The home-built LLS-SIM system was developed from the original design[36]. Similar to the laser combinator and pattern modulator used in our Multi-SIM system, three lasers of 488 nm, 560 nm, and 640 nm (MPB Communications) were selected and controlled by an AOTF, and then modulated by the lattice patterns displayed on the SLM. The excitation light was then filtered by an annular mask equivalent to 0.5 outer NA and 0.375 inner NA for the excitation objective (Special Optics). Subsequently, the filtered excitation light passed through a pair of galvo mirrors (x- and z-galvo) (Cambridge Technology, 6210H). In LLS-SIM mode, the lattice patterns of 3-phase were sequentially displayed on the SLM and synchronized with the programmed "ON" time of AOTF, and then scanned by the sample piezo in a step size of 0.39 μm, which equals to a z-interval of 0.2 μm, to acquire the volumetric LLSM images. In LLSM mode, a fixed lattice pattern was quickly dithered by x-galvo, and then scanned by the sample piezo. In particular, we used the triangle wave when reversing the scanning direction of the piezo stage to minimize the flyback time to an extreme. Live cell specimens were held in a customized microscope incubator (OKO Lab, H301-LLSM-SS316) to maintain the physiology condition of 37°C and 5% $CO_2$ during imaging. The emission light was collected by the detection objective (Nikon, CFI Apo LWD 25XW, 1.1NA) and captured by a sCMOS camera (Hamamatsu, Orca Fusion).

## Confocal system

The home-built confocal microscopy was developed as a modification of the image-scanning microscopy system[60] based on a commercial invented fluorescence microscope (Ti2E, Nikon). Four laser beams of 405 nm, 488 nm, 561 nm, and 640 nm (BDL-405-SMN, BDL-488-SMN, BDS-561-SMY-FBE, and BDL-640-SMN, Becker & Hickel) were collinearly combined and then expended by 6.25 times. After being reflected by a multi-band dichroic mirror (Di03-R405/488/561/635, Semrock), the lasers were passed through two galvanometer scanners (8315k, CT Cambridge Technology) and then directed toward the objective (CFI SR HP Plan Apo Lambda S 100XC/1.35NA, Sil, Nikon) via a scan lens and a tube lens. The emission fluorescence was collected by the same objective, descanned, and passed through the multi-band dichroic mirror and then separated into the green channel and then red channel by a dichroic beam splitter (FF573-DI01, Semrock). The green-channel signals (filtered by FF02-447/60, FF03-525/50, Semrock) were collected by a single photon counting module (SPCM-AQRH-44, Excelitas) and finally counted by a digital counter (BNC-2121, National Instruments). The red-channel signals (filtered by FF01-609/57, FF01-679/41, Semrock) were collected by a fiber bundle and then captured by a multi-channel photomultiplier tube (PML-16-GASP) and quantified by a single photon counter (SPC-164-PCI, Becker & Hickel). The pinhole was kept open during image acquisition and the overall magnification factor was 333× for the green channel and 666× for the red channel. The data acquisition/visualization/processing was operated by a home-developed software based on LabView (National Instruments) and the software also controlled all microscope devices during the image acquisition, such as the galvanometer scanners, the axial piezo stage, and the laser power by sending analog signals via a field-programmable-gate-array card (NI PXIe-7868R, National Instruments).

## Architectures and objective functions of ZS-DeconvNet

ZS-DeconvNet adopts a dual-stage architecture, which factorizes low-SNR super-resolution task into two sequential subdivisions of denoising and deconvolution, and each stage is responsible for one subtask, respectively. The dual-stage design is helpful to regulating the training procedures and eliminating the noise-induced artifacts in the final outputs[11]. For 2D images, a simplified U-Net model[29] with four down-and up-sampling modules is used as the backbone of each stage. The overall network architecture of ZS-DeconvNet we used for 2D image SR in this work is shown in Supplementary Fig. 2a. In the training phase, we designed a combined loss function consisting of a denoising term and a deconvolution term, which respectively corresponds to the denoising stage and the deconvolution stage:

$$\mathcal{L}(\hat{\mathbf{y}}, \widetilde{\mathbf{y}}) = \mu \mathcal{L}_{den}(\hat{\mathbf{y}}, \widetilde{\mathbf{y}}) + (1-\mu)\mathcal{L}_{dec}(\hat{\mathbf{y}}, \widetilde{\mathbf{y}}) \tag{3}$$

where $(\hat{\mathbf{y}}, \widetilde{\mathbf{y}})$ indicates the recorrupted image pair (see next section for the details of image recorruption), and $\mu$ is a scalar weighting factor to balance the two terms, which we empirically set as 0.5 in our experiments. We have also validated that the performance of ZS-DeconvNet is stable on all the samples for a large range of $\mu$ (Supplementary Fig. 29). The denoising loss $\mathcal{L}_{den}(\hat{\mathbf{y}}, \widetilde{\mathbf{y}})$ and the deconvolution loss $\mathcal{L}_{dec}(\hat{\mathbf{y}}, \widetilde{\mathbf{y}})$ are defined as follows:

$$\mathcal{L}_{den}(\hat{\mathbf{y}}, \widetilde{\mathbf{y}}) = \left\| f_{\boldsymbol{\theta}'}(\hat{\mathbf{y}}) - \widetilde{\mathbf{y}} \right\|_2^2 \tag{4}$$

$$\mathcal{L}_{dec}(\hat{\mathbf{y}}, \widetilde{\mathbf{y}}) = \left\| (f_{\boldsymbol{\theta}}(\hat{\mathbf{y}}) * \text{PSF})_{\downarrow} - \widetilde{\mathbf{y}} \right\|_2^2 + \lambda \mathcal{R}_{Hessian}(f_{\boldsymbol{\theta}}(\hat{\mathbf{y}})) \tag{5}$$

where PSF denotes the point spread function of the optical system, $(\cdot)_{\downarrow}$ is the down-sampling operator, $f_{\boldsymbol{\theta}'}(\hat{\mathbf{y}})$ and $f_{\boldsymbol{\theta}}(\hat{\mathbf{y}})$ are the output images of the denoising stage and the deconvolution stage, $\mathcal{R}_{Hessian}(\cdot)$ is the Hessian regularization term used to regulate the solution space, and $\lambda$ is the weighting scalar to balance the impact of the regularization, which we empirically set as 0.02 for the best performance in implementations of 2D ZS-DeconvNet.

For 3D ZS-DeconvNet, we deploy 3D RCAN as the backbone model for the two stages, each of which includes two residual groups consisting of two channel attention blocks. The overall architecture is illustrated in Supplementary Fig. 2b. During training procedures, the 3D ZS-DeconvNet is optimized iteratively following a similar loss function to its 2D versions, nevertheless, with two major modifications in detail: first, the image pairs used for training were generated by axial sampling rather than via recorruption, resulting in a totally parameter-free data augmentation strategy; second, the gap amending regularization (GAR)[9] was implemented in both denoising term and deconvolution term to correct the inconsistency between the inputs and targets which are originally interleaved in the same noisy image stack. The loss function can be formulated as:

$$\mathcal{L}(\mathbf{z}) = \mu \mathcal{L}_{den}(\mathbf{z}) + (1-\mu)\mathcal{L}_{dec}(\mathbf{z})$$
$$\mathcal{L}_{den}(\mathbf{z}) = \left\| f_{\boldsymbol{\theta}'}(S_{odd}(\mathbf{z})) - S_{even}(\mathbf{z}) \right\|_2^2 \tag{6}$$

$$+ \gamma \left\| f_{\boldsymbol{\theta}'}(S_{odd}(\mathbf{z})) - S_{even}(\mathbf{z}) - (S_{odd}(f_{\boldsymbol{\theta}'}(\mathbf{z})) - S_{even}(f_{\boldsymbol{\theta}'}(\mathbf{z}))) \right\|_2^2 \tag{7}$$

$$\mathcal{L}_{dec}(\mathbf{z}) = \left\| (f_{\boldsymbol{\theta}}(S_{odd}(\mathbf{z})) * \text{PSF})_{\downarrow} - S_{even}(\mathbf{z}) \right\|_2^2$$

$$+ \gamma \left\| (f_{\boldsymbol{\theta}}(S_{odd}(\mathbf{z})) * \text{PSF})_{\downarrow} - S_{even}(\mathbf{z}) - S_{odd}(f_{\boldsymbol{\theta}'}(\mathbf{z})) + S_{even}(f_{\boldsymbol{\theta}'}(\mathbf{z})) \right\|_2^2$$
$$+ \lambda \mathcal{R}_{Hessian}(f_{\boldsymbol{\theta}}(S_{odd}(\mathbf{z}))) \tag{8}$$

where $\mathbf{z}$ is the 3D noisy image stack, $S_{odd}(\cdot)$ and $S_{even}(\cdot)$ represent the axial sampling operator which takes an image stack and returns its odd slices or even slices, respectively, stacked in the same order as the original stack, $\gamma$ and $\lambda$ are weighting scalars of the GAR term and the Hessian regularization term, which are set to $\gamma = 1$, $\lambda = 0.1$ for the implementation of 3D ZS-DeconvNet.

It is noteworthy that since the theoretical basis of ZS-DeconvNet is model-agnostic, both U-Net and RCAN are not the only applicative backbone models but the widely adopted and efficient ones. Equipping ZS-DeconvNet with other state-of-the-art network architectures, e.g., DFCAN[8] and RLN[12], may further improve its denoising and SR capability.

## Implementation of 2D ZS-DeconvNet

The image pairs $(\hat{\mathbf{y}}, \widetilde{\mathbf{y}})$ used for training 2D ZS-DeconvNet models were generated following a modified scheme from the original recorrupted to recorrupted strategy[26] under the assumption of mixed Poisson-Gaussian noise distributions, where three hyperparameters $\beta_1$, $\beta_2$, $\alpha$ needed to be pre-characterized. The recorruption procedure from a single noisy image $y$ can be represented in matrix form as:

$$\hat{\mathbf{y}} = \mathbf{y} + D\mathbf{g} \qquad (9)$$

$$\widetilde{\mathbf{y}} = \mathbf{y} - D^{-1}\mathbf{g} \qquad (10)$$

where $D = \alpha I$ is an invertible matrix defined as a magnified unit matrix by a factor of $\alpha$, which controls the overall magnitude of added noises, and $\mathbf{g}$ is a random noise map sampled from a Gaussian distribution with zero means:

$$\mathbf{g} \sim \mathcal{N}(0, \sigma^2 I) \qquad (11)$$

$$\sigma^2 = \beta_1 H(\mathbf{y} - \mathbf{b}) + \beta_2 \qquad (12)$$

where $\beta_1$ is the Poissonian factor affecting the variance of the signal-dependent shot noise, and $\beta_2$ is the Gaussian factor representing the variance of additive Gaussian noises. $\mathbf{b}$ is the background, approximately regarded as a fixed value related to the camera, by subtracting which we extracted fluorescence signals from the sample. $H(\cdot)$ is a linear low-pass filter used to preliminarily smooth the image and reduce the noise, and we adopted an averaging filter with a size of 5 pixels in our experiments.

As is proved in Supplementary Note 1, the theoretically optimal value of both $\beta_1$ and $\alpha$ is 1, while $\beta_2$ is dependent to the camera and can be estimated from the sample-free region of the image itself or pre-calibrated following standard protocols[61]. Evaluations on simulated data has shown that the best denoising and SR performance are achieved at the theoretically optimal values of these hyperparameters regardless of the structure and SNR of the testing images (Supplementary Figs. 3, 4).

## Implementation of 3D ZS-DeconvNet

The training scheme of 3D ZS-DeconvNet integrates the spatially interleaved self-supervised learning scheme[9] with the self-supervised inverse problem solver. In the training process, each noisy image stack was divided into odd slices and even slices, which were then used as input and targets, respectively, after augmentation by random rotating, cropping, and flipping. To amend the expectation gap between odd and even slices, we introduced the gap amending regularization (GAR) term into both denoising loss and deconvolution loss, which was calculated with the denoised stack (labeled with the red box in Fig. 3a), noisy even slices, and network outputs (detailed in Supplementary Note 1b).

## Implementation of 2D/3D ZS-DeconvNet-SIM

For ZS-DeconvNet-SIM implementations on 2D-SIM and 3D-SIM, every set of raw SIM images were first augmented into two sets of recorrupted raw images through Eq. 9 and 10, and reconstructed into a pair of SR SIM images via the conventional SIM reconstruction algorithm.

The generated SIM image pairs were then used for self-supervised training in a similar manner to training the ZS-DeconvNet models. For 3D ZS-DeconvNet-SIM applied on LLS-SIM (Fig. 5d, e), post-reconstructed volumetric SIM data instead of the raw images were axially sampled into two SIM stacks respectively containing odd and even slices, which were used in subsequent training procedures of 3D ZS-DeconvNet models with loss functions described in Eq. 6-8. The schematic workflow of ZS-DeconvNet-SIM is shown in Fig. 5a and Supplementary Fig. 20.

## PSF usage and generation

In the training procedure of ZS-DeconvNet, we used experimentally acquired or simulated PSFs (with PSF Generator Fiji plugin licensed by EPFL) that are corresponding to the imaging configurations. Independent ZS-DeconvNet models were trained for each biological structure and emission wavelength for best performance.

## Model training and test-time adaptation

In this work, ZS-DeconvNet models were trained on a PC with an Intel Core i7-11700 processor and an RTX 3090 graphic processing card (NVIDIA) under the software environment of TensorFlow 2.5.0 and python 3.9.7. Before training, the paired input/GT images were first augmented into several patch pairs via random cropping, horizontal/vertical flipping and rotation transformation to further enrich the training dataset, which eventually generated ~20,000 pairs of 2D patches ($128 \times 128$ pixels) or ~10,000 pairs of 3D patches ($64 \times 64 \times 13$ voxels). Training was typically conducted with the Adam optimizer and an initial learning rate of $0.5 \times 10^{-4}$, which would decay with a factor of 0.5 every 10,000 iterations. Training batch size was 4 for 2D images and 3 for 3D stacks. The entire training process usually required 50,000 iterations for 2D images and 10,000 iterations for 3D stacks. Elapsed time of training 50,000 iterations for 2D models and 10,000 iterations for 3D models was ~1 h and ~2 h, respectively. As is often the case with most deep learning-based methods, the training of ZS-DeconvNet is a one-time procedure in most live-cell imaging cases, where users train the ZS-DeconvNet model with all frames, then the well-trained models are applicable for all data of the same biological specimen at a high processing speed. To eliminate the edge artifacts induced by deconvolution, we typically padded 2 blank slices at the top and bottom of 3D stacks and a margin of 8 pixels for each xy-slice in both training and inference processes (Supplementary Fig. 30a). Particularly, when processing the time-lapsing data of cell mitosis (Fig. 3e, f), the unsupervised property of ZS-DeconvNet enabled a test-time adaptation learning strategy[43] in which we first trained a general model for each biological structure with data of the entire process and then finetuned the pre-trained model for each timepoint with a small number of training steps (typically 50 iterations taking ~1 min) to fully exploit the structural information of the raw data and obtain the optimal SR performance. Of note, the test-time adaptation is not necessary but an optional technique to improve the performance of ZS-DeconvNet especially under circumstances where there are huge morphological changes on biological specimens during the observation window, e.g., the chromosomes during mitosis (Supplementary Fig. 31).

## Data post-processing and SR image evaluation

For imaging modalities employing wide-filed detection such as LLSM, the fixed pattern noise (FPN) which are induced by the nonuniformity in the pixel sensitivity of the camera cannot be removed by noise2noise-based schemes[62]. In our implementation of ZS-DeconvNet, the FPN would be enhanced in the deconvolution stage and became nonnegligible especially at imaging conditions of extremely low SNR. For sCMOS sensors, which are the most common in fluorescence microscopy, the fixed pattern usually presents a regular appearance of horizontal or vertical stripes attributed to the column

amplifier. To this end, we simply applied an apodization mask in Fourier domain to suppress the stripy artifacts while preserving other frequency components of the samples (Supplementary Fig. 30b). It is noted that the fixed pattern noise can also be fundamentally removed by pre-calibration for the acquired raw images before sent into the network model following the well-established procedures[61,63,64].

Other computational SR approaches compared in this work, i.e., the sparse deconvolution[5], DeepCAD-based deconvolution[33], and SRRF[13] are implemented following the instructions in the original papers. Specifically, we tried our best to select the optimal hyperparameters for sparse deconvolution to obtain a reconstructed image with the least artifacts and the highest resolution. And the DeepCAD-based deconvolution (Figs. 2a and 3f) was carried out by integrating the temporally sampling scheme into our ZS-DeconvNet framework, that was, using images temporally sampled from the time-lapsing data for training our dual-stage network models, ensuring the same model size and computational cost for a fair comparison.

To quantitatively evaluate the SR performance of 2D ZS-DeconvNet and other computational SR approaches with only diffraction limited references, we calculated PSNR between clear WF targets and SR images degraded with the PSF by following three steps: (1) Convolving the SR image with the corresponding PSF and downsampling the convolved image $\mathbf{I}$ to the size of GT; (2) Normalizing the GT image $\mathbf{x}$ to the range of [0, 1] and then applying a linear transformation[8,53] to the convolved image $\mathbf{I}$ to match its dynamic range with $\mathbf{x}$:

$$\mathbf{I}_{trans} = a\mathbf{I} + b \tag{13}$$

$$(a,b) = argmin_{(\theta_1,\theta_2) \in \mathbb{R}^2} \left( ||\theta_1\mathbf{I} + \theta_2 - \mathbf{x}||_2^2 \right) \tag{14}$$

The linear transformation is applied to all methods for a fair comparison; (3) Calculating the PSNR between the normalized GT image $\mathbf{x}$ and linearly transformed image $\mathbf{I}_{trans}$.

For PSNR evaluation of 3D ZS-DeconvNet (Fig. 3d), we directly leveraged the LLS-SIM images as the reference in that both LLS-SIM and our 3D ZS-DeconvNet provided a resolution improvement by ~1.5-fold theoretically. The overall calculation process is similar to the 2D cases, except that the SR stacks were not convolved and the PSNR was only calculated within the feature-only regions with a threshold of 0.02 to avoid obtaining an abnormally high value of PSNR.

To provide better contrast and visualization, we equally performed percentile normalization for the deconvolution images generated by RL deconvolution, sparse deconvolution, and ZS-DeconvNet, which is formulated as:

$$\text{Norm}_p\left(\mathbf{Y},p_{low},p_{high}\right) = \frac{\mathbf{Y} - \text{percentile}(\mathbf{Y},p_{low})}{\text{percentile}\left(\mathbf{Y},p_{high}\right) - \text{percentile}\left(\mathbf{Y},p_{low}\right)}, \tag{15}$$

where percentile($\mathbf{Y}$,$p$) outputs the intensity value ranking $p$% in image $\mathbf{Y}$. $p_{low}$ and $p_{high}$ are typically set as 3 and 100 in our figure and videos.

## Cell culture, transfection, and staining

Cos7, HeLa, 293 T cells as well as their stable cell lines were cultured in DMEM (Gibco, cat. no. 11965092), supplemented with 10% fetal bovine serum (Gibco, cat. no. 10099141 C) and 1× penicillin-streptomycin (Thermo Fisher, 15140122) under 37°C in Thermo Scientific™ Heracell™ 150i CO$_2$ incubator. SUM159 cells were cultured in DMEM/F12K medium supplementary with 5% Fetal Bovine Serum (FBS) and 1% Penicillin-Streptomycin solution.

For live cell imaging, the 35 mm coverslips were pre-coated with 50 μg ml$^{-1}$ of collagen and 1×10$^5$ cells were seeded onto coverslips. For transient transfection, cells were transfected with plasmids using Lipofectamine 3000 (Invitrogen, cat. no. L3000150) according to the manufacturer's protocol 12 h post plating. Cells were imaged for 12 h after transfection. Where indicated, the cells transfected with Halo Tag plasmids were labeled with 10 nM JF549 ligand for 15 min according to the published protocol[65]. The cells were rinsed with fresh medium to remove unbound ligand and imaged immediately afterward. The plasmids used in transient transfection include Lifeact-mEmerald, Clathrin-mEmerald, 3×mEmerald-Ensconsin, Lamp1-Halo, 2×mEmerald-Tomm20, Myosin2-Halo, KDEL-mCherry, and Halo-Calnexin.

For lentivirus packaging, 1 μg lentiviral transfer vector DNA, together with 0.5 μg psPAX2 packaging and 0.5 μg pMD2.G envelope plasmid DNA were co-transfected to 90% confluence HEK293T cells in a 6 cm petri dish using Lipofectamine 3000 following the manufacturer's protocol. After 2 days, supernatant was harvested and filtered with a 0.22-μm filter (Millipore). For construction of stable cells, HeLa and Cos7 cells were infected with lentiviruses encoding endoplasmic reticulum marker Calnexin-mEmerald and F-actin marker Lifeact-mEmerald[66]. Forty-eight hours after, the cells were enriched by flow cytometer (FACSAria III, BD Biosciences) and then plated one cell per well into 96-well plates, Monoclonal cells were used for our experiments. Specifically, Lifeact-mEmerald for COS7 used in Figs. 3 and 5; Calnexin-mEmerald, Mito-dsRed and Halo-H2B for HeLa cells used in Fig. 3; H2B-mCherry for HeLa-mEmerald-SC35 used in Supplementary Fig. 18.

## Genome edited cell lines

SUM159 cells were genome edited sequentially to incorporate EGFP to the N-terminus of Rab11A and then Halo to the C-terminus of Lamp1 using the CRISPR/Cas9 approach[67,68]. The single-guide RNA (sgRNA) targeting sequences are 5'-TCGCTCCTCGGCCGCGCAAT-3' for RAB11A and 5'-CTATCTAGCCTGGTGCACGC-3' for LAMP1. SUM159 were transfected with the EGFP-Rab11A donor plasmid, the plasmid coding for the spCas9 and the free PCR product containing the sgRNA targeting sequence using Lipofectamin 3000 (Invitrogen) according to the manufacturer's instruction. The cells expressing EGFP were enriched by fluorescence-activated cell sorting (FACS) (FACSAria II, BD Biosciences), and further subjected to single cell sorting to 96-well plates. The monoclonal cells with successful EGFP incorporation were identified by PCR screening using GoTaq Polymerase (Promega). The clonal SUM159 cells expressing EGFP-Rab11A +/+ were subjected to the second round of genome editing to incorporate Lamp1-Halo in the genome as described above. The transfected cells were stained by 10 nM Janelia Fluor 646 HaloTag Ligands (Promega) for 15 min. To wash the unbound dye, samples were rinsed with fresh medium, and then enriched by FACS. The monoclonal SUM159 cells expressing both EGFP-Rab11A +/+ and Lamp1-Halo +/+ were confirmed by PCR and Western blot analysis.

SUM159 cells were genome edited to incorporate EGFP to the C-terminus of clathrin light chain A (clathrin-EGFP) using the TALEN-based approach[69]. The clathrin-EGFP expressing cells were enriched by two sequential bulk sorting.

HeLa cells lines were genome edited to incorporate mEmerald into the C- terminus of human genomic SC35 using CRISPR-Cas9 gene editing system. The sgRNA targeting sequence is 5'-CGAGCAGCACTCCTAATGAT-3', and the sgRNA was ligated into pX330A-1×2 (Addgene, 58766). The resulting plasmid was named pX330-SC35-gRNA hereafter. To construct donor vector p-SC35-doner, mEmerald flanked with about 1800bp homology arms complementary to the stop codon of human genomic SC35 locus were ligated to pEASY-blunt (Transgene, CB101). 2×10$^5$ HeLa cells grown in 6 cm petri dish were transfected with 1.2 μg of pX330-SC35-gRNA and 0.4 μg of p-SC35-doner. 48 h post transfection, mEmerald-positive cells were sorted

using FACS (FACSAria III, BD Biosciences). After one week, H2B-mCherry lentivirus were infected sorted cells and then single cells were seeded into 96-well. After two weeks, genomic DNA of different single cell clones were extracted and validated by PCR and western blot. Homozygous SC35 knock-in cells were selected for the study. The successful SC35 knock-in was verified by PCR and Western blot analysis.

### *C. elegans* embryo preparation

*C. elegans* strains were cultured at 20 °C on nematode growth medium (NGM) plates seeded with OP50 following standard protocols[70]. TV52712*[wyEx51119[dlg-1p::GFP::PLCdPH]*; *jcIs1[ajm-1::GFP + UNC-29(+) +rol-6(su1006)]*; *qxIs257 [ced-1p::nuc-1::mCherry + unc-76(+)]]* was used in this study. The plasmid *dlg-1p::GFP::PLCdPH* was constructed following the Clontech In-Fusion PCR Cloning System[71] and microinjected to *jcIs1;qxIs257*. Extrachromosomal array *wyEx51119* marked epidermal cell membrane. *jcIs1* marked the apical junctional domain of *C. elegans*[71]. *qxIs257* marked lysosomes in epidermal cells[72].

About 50 L4 stage transgenic worms were put onto NGM plates with freshly OP50 48 to 60 h before experiments. Transgenic eggs were collected under the dissecting fluorescent microscope (Olympus MVX10), and mounted on 3% agarose pads. Lima bean to 2-fold stage embryos were then imaged using the 3D WF mode of our Multi-SIM system.

### Mouse embryo preparation

Mice used in this study were of C57BL/6 J background. All animal experiments were approved by the Animal Care and Use Committees (IACUC) of the Institute of Biophysics, Chinese Academy of Sciences, Beijing, China. Pre-implantation embryos were isolated from 5-6-week-old females, superovulated by intraperitoneal injection of 5 international units (IU) of pregnant mares' serum gonadotropin (PMSG; LEE BIOSOLUTIONS) and 5 IU human chorionic gonadotropin (hCG; Millipore) 48 h later, and mated with male mice. Zygotes were recovered at E0.5 in M2 medium (Millipore) and cultured in KSOM medium (Millipore) in $CO_2$ incubator (Thermo Scientific) at 37°C with 5% $CO_2$ until the late 8-cell stage.

For immunofluorescence, embryos were fixed with 4% paraformaldehyde in PBS for 30 min at room temperature (RT) and washed with PBS three times. Embryos were then permeabilized in 0.5% TritonX-100 (Sigma) in PBS for 20 min at RT, washed in PBS three times, blocked in 1% bovine serum albumin in PBS for 1 h at RT and incubated with anti-pERM antibody (Abcam, ab76247), anti-alpha-tubulin-FITC (Sigma, F2168-.2 ML) and Phalloidin-Rhodamine (Molecular Probes, R415) overnight at 4°C. Then, embryos were washed in PBS three times, incubated with secondary antibodies (Life technologies) for 1 h at RT, stained with Hoescht 33342 (Thermo) for 15 min at RT, washed in PBS three times and imaged by the home-built confocal microscope.

### 3D image visualization

The axially color-coded images of lysosomes shown in Fig. 4f, g were generated with Fiji. The 3D rendering images of mitosis cell and mouse embryos shown in Fig. 3e, f were visualized and generated by using of the commercial software Amira.

### Statistics and reproducibility

Experiments in Figs. 2a–i, 3f, 4a–h, and 5b–e were independently repeated with at least 3 specimens, i.e., cells or embryos, all achieving similar results.

### Reporting summary

Further information on research design is available in the Nature Portfolio Reporting Summary linked to this article.

## Data availability

The SIM data of CCPs and MTs used for evaluating ZS-DeconvNet is from the publicly accessible dataset BioSR (https://doi.org/10.6084/m9.figshare.13264793). Other data that are generated and presented in Figs. 1–5, Supplementary Figs. 1-34, and Supplementary Videos 1–9 in this study are available upon requests. Source data are provided with this paper.

## Code availability

The python codes of ZS-DeconvNet, the Fiji plugin, several representative pre-trained models, as well as some example data for training and testing are already publicly accessible on the tutorial homepage (https://tristazeng.github.io/ZS-DeconvNet-page/) of ZS-DeconvNet and Github repository[73] (https://github.com/TristaZeng/ZS-DeconvNet).

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

## Acknowledgements

The authors thank T. Kirchhausen for the donor plasmids used for genome editing and help in generating the genome-edited cell lines, and thank the Prof. Xiaochen Wang and Dr. Kangmin He for *C. elegans* strains and genome-edited SUM159 cell lines. This work was supported by grants from the National Natural Science Foundation of China (32125024, 32271513, 62071271, and 62088102); the Ministry of Science and Technology (2021YFA1300303 and 2020AA0105500); the Chinese Academy of Sciences (ZDBS-LY-SM004 and XDA16021401); the Collaborative Research Fund of the Chinese Institute for Brain Research, Beijing (2021-NKX-XM-03); China Postdoctoral Science Foundation (2022M721842, 2023T160365); the New Cornerstone Science Foundation; the Shuimu Tsinghua Scholar Program (2022SM035); Beijing Natural Science Foundation (JQ21012).

## Author contributions

Q.D. and Dong Li supervised the research. Q.D., Dong Li, and C.Q. conceived and initiated this project. C.Q. designed the detailed implementations under the instruction of Q.D. and Dong Li. Y.Z, C.Q., and X.C developed the python code, performed simulations, and processed relevant imaging data. H.C., C.Q., and Y.Z. developed the Fiji plugin. T.J., R.W, C.Q, H.L., W.F., Di Li, and J.G. prepared samples and performed imaging experiments. C.Q., Y.Z., X.C., and Q.M. analyzed the data with conceptual advice from Q.D., Dong Li, J.W, Y.W., and H.Q. C.Q, Y.Z, and Q.M. composed the figures and videos, made the tutorial homepage under the supervision of Q.D. and Dong Li. Q.D., Dong Li, and C.Q. wrote the manuscript, with input from all authors. All authors discussed the results and commented on the manuscript.

## Competing interests

Dong Li, C.Q. and Y.Z. filed a patent as inventors through Institute of Biophysics, Chinese Academy of Sciences, to the Chinese Patent Office (Pub. No. CN116721017A & App. No. 202310735660.3), which contains the basic application of the presented ZS-DeconvNet framework. The remaining authors declare no competing interests.
