## [Peer Review File · Nature Communications]

Zero-shot learning enables instant denoising and super-resolution in optical fluorescence microscopyEditorial Notes: This manuscript has been previously reviewed at another journal that is not operating a transparent peer review scheme. This document only contains reviewer comments and rebuttal letters for versions considered at *Nature Communications*. Parts of this Peer Review File have been redacted as indicated to remove third-party material where no permission to publish could be obtained.

REVIEWER COMMENTS

Reviewer #1 (Remarks to the Author):

The authors have made a commendable effort to address my previous comments and suggestions (reviewer #1) as well as those of the other reviewers, resulting in a further improvement manuscript. My general assessment has not changed, in that I find this work of outstanding quality and that the introduction of ZS-DeconvNet is a significant achievement for the general applicability of deep learning-based microscopy data post-processing for denoising and resolution enhancement.

One minor comment is that the authors should cite and discuss more recent publications in the field, notably eSRRF (Laine et al. 2023, Nat Methods) and DL-axial resolution improvement in 3D-SIM (Li & Wu et al., Nat Biotech 2023).

Wording: Line 46/47: "..., enabling significant upgradation of existing FM systems and extension of their applicable range." should read "enabling a significant upgrade of existing FM systems and extension of their application range."

Line 53: "and prone" -> "and are prone".

Reviewer #2 (Remarks to the Author):

Summary: Rejected for nature communication.

Novelty & significance:

The manuscript mentions existing supervised deep learning models for SR but lacks a clear comparison with the proposed unsupervised approach. How does ZS-DeconvNet's

performance compare to state-of-the-art supervised models in terms of resolution enhancement, image quality, and generalization across modalities? Do we need quantification results on the larger microscopy datasets? The authors didn't mention why RL-based deconvolution is the comparison method when assessing the reconstruction data quality.

Methodology:

The manuscript uses a two-stage network but doesn't delve into its uniqueness. From the methodology, the authors proposed simple image reconstruction networks for the tasks as denoising and deconvolution, which are already a proposed models (example: <https://opg.optica.org/prj/fulltext.cfm?uri=prj-9-5-B168&id=450235>) and using multiple networks such as U-net for denoising and U-net for deconvolution process, or RCAN networks for the deconvolution process is not a unique contribution. Quantification of the reported 1.5x resolution enhancement and 10x lower fluorescence power should be reported on a large dataset?

Usability:

Missing a table to show what pretrained model to use for a specific application? See the attached image during the inference time.

I missed a table to show the list of all trained networks for each sample or modality for 2D and 3D samples, including their training parameters (learning rate, epochs, steps, loss, time for each epoch, results of the improvement in the case of denoising and super-resolution, PSNR, SSIM, and FRC values).

It is not easy for a common user to understand how to use this FIJI application tool and get quick results within due time to verify whether these models are actually helpful or not.

Additional comments:

The work described in the manuscript involves the ZS-deconvolution application on microscopy techniques and genome editing methods to study cellular and developmental biology in different model systems, including cell lines, *C. elegans* embryos, and mouse embryos. Key aspects of the work include:

There are several additional aspects of the paper that may warrant critical scrutiny and

further clarification:

1. Justify the choice of specific percentiles for image normalization and provide clarity on how they were determined.
2. Enhance clarity in describing cell culture, transfection, and staining protocols, including rationale and references for each step.
3. Provide details on sgRNA/TALEN design, efficiency, and validation methods to ensure specificity and reliability of genome editing.
4. Include specific details on imaging parameters and analysis methods to assess data quality and validity accurately.
5. Expand on why specific methods were chosen (e.g., Lipofectamine 3000 vs. other transfection reagents). Justify any non-standard techniques used.
6. Discuss potential off-target effects of genome editing tools and strategies for minimizing or evaluating their impact on experimental outcomes.
7. Provide comprehensive information about reagents, materials, and sources used in experiments to facilitate reproducibility and transparency.
8. Provide comprehensive information about reagents, materials, and sources used in experiments to facilitate reproducibility and transparency.
9. Include details on statistical methods used for data analysis, including sample size determination, significance thresholds, and appropriate tests for comparisons.
10. Include specifics like microscope settings, objective lenses, resolution, and exposure times. This allows for replication and evaluation of image quality.
11. Describe any image processing steps applied (e.g., filtering, background subtraction). Mention software used and relevant parameters.
12. Briefly outline future research directions based on your findings. This demonstrates the broader significance of your work and stimulates further investigation.
13. Include the following state-of-the-art methods relevant to the study: Noise2Noise imageJ plugin for image denoising, RL-based deconvolution for the fluorescence microscopy. There are multiple studies shows the effect of small training datasets for the super-resolution using DeepSTORM, application specific super-resolutions methods to show the minimum dataset required for the SR training methods.
14. Missing for 2PE/3PE fluorescence microscopy imaging to show the proposed method works on the highly scattering tissues?

15. Missing the PSF generation method for each modality and how to handle this PSF phenomenon during the inference stage?

16. Missing information about the disrupting the noisy images? What is the need, what type of noise is added, what is the distribution of noise added, and what are the PSNR values?

Overall, this proposed method works only you have a SR microscopy setup which is an expensive and not a common optical fluorescence bench setup for in vivo imaging.

RESPONSE TO REVIEWERS

We thank the reviewers for their constructive suggestions, which have substantially improved our manuscript. We believe we have addressed all of the major and minor comments. Changes made to the manuscript are summarized in the point-by-point response to each comment. The revised and added texts have been highlighted in the manuscript and supplementary materials. All numerical references to figures, tables, or citations in the manuscript or supplementary information refer to the numbering in the newly submitted version.

REVIEWER COMMENTS

Reviewer #1:

The authors have made a commendable effort to address my previous comments and suggestions (reviewer #1) as well as those of the other reviewers, resulting in a further improvement manuscript. My general assessment has not changed, in that I find this work of outstanding quality and that the introduction of ZS-DeconvNet is a significant achievement for the general applicability of deep learning-based microscopy data post-processing for denoising and resolution enhancement.

We sincerely appreciate these positive comments by recognizing the key merits of our work.

1. One minor comment is that the authors should cite and discuss more recent publications in the field, notably eSRRF (Laine et al. 2023, Nat Methods) and DL-axial resolution improvement in 3D-SIM (Li & Wu et al., Nat Biotech 2023).

Appreciate this suggestion. We have made comparisons between ZS-DeconvNet and five recently reported representative methods for *denoising* (DeepSeMi: Zhang, G. et al. Nat. Methods 20, 1957-1970, 2023; SRDTrans: Li, X. et al. Nat. Computational Science 3, 1067-1080, 2023), *super-resolution* (eSRRF: Laine, R. F. et al. Nat. Methods 20, 1949-1956, 2023), and self-supervised *axial resolution enhancement* (DL-ARE: Li, X. et al. Nat. Biotechnology, 2023; Self-Net: Ning, K. et al. Light: S & A, 12, 204, 2023) of biological images in Response Fig. 1 and Supplementary Fig. 13 and 17. Specifically, we demonstrate:

(i) Compared with analytical model-based computational SR methods such as eSRRF, ZS-DeconvNet presents stronger noise-robustness and better output fidelity in the single image denoising and SR task (Response Fig. 1).

Response Fig. 1

Response Fig. 1 | Performance comparison of eSRRF and ZS-DeconvNet on the single image SR task.

It is noteworthy that eSRRF (as well as its previous version SRRF) is a fluctuation-based SR approach, which in theory requires several frames of the same scene with frame-to-frame intensity

variations to reconstruct a single SR image. Although its codes and Fiji plugin can be executed with one frame input, eSRRF is originally developed for a different application scenario from ZS-DeconvNet. Therefore, we only show results in the above response figure but not a supplementary figure for fair comparison.

(ii) Compared with state-of-the-art self-supervised denoising methods such as the most recent SRDTrans and DeepSeMi, ZS-DeconvNet not only provides a comparable denoised result (magnified region #6 v.s. regions #4 and #5) but also generates a super-resolved image (magnified region #3) as is shown in the new Supplementary Fig. 15, indicating its more versatile functionality.

Supplementary Fig. 15

Supplementary Fig. 15 | Comparisons of ZS-DeconvNet with self-supervised denoising methods. Images of mitochondrial outer membrane captured or processed by LLSM (a for low SNR, b for high SNR), ZS-DeconvNet (c for deconvolved results, f for denoised results), SRDTrans (d), and DeepSeMi (e). Single-slice images of the region labelled by the red box and magnified regions labelled by yellow boxes are shown for better comparison. Scale bars, 5 μm (MIP), 3 μm (single slice), and 1 μm (magnified region).

(iii) Compared with dedicated DL-based axial resolution enhancement techniques such as DL-ARE and Self-Net, although ZS-DeconvNet could not achieve as high as more than 2-fold axial resolution enhancement, it provides a modest 1.5 \times resolution improvement beyond the diffraction limit along all dimensions (Fig. 3c, d, and Supplementary Fig. 17f). Notably, due to the severe photon noise, neither DL-ARE nor Self-Net could effectively improve the axial resolution from noisy raw LLSM images (Supplementary Fig. 17c, d, i), however, ZS-DeconvNet can serve as a preprocess for subsequent DL-ARE or Self-Net to enable stronger noise-suppression capability and higher final axial

resolution (Supplementary Fig. 17g-i).

Supplementary Fig. 17

Supplementary Fig. 17 | Comparison of ZS-DeconvNet and other methods in axial resolution improvement. **a**, Representative noisy LLSM images (max intensity projection, MIP) of ER (left), F-actin (middle), and mitochondria (right). **b-h**, XZ-slices, of which the Y-positions are indicated by yellow dashed lines in **a**, from corresponding image stacks without processing (**b**) and processed by Self-Net (**c**), DL-ARE (**d**), sparse deconvolution (**e**), ZS-DeconvNet (**f**), ZS-DeconvNet + Self-Net (**g**), and ZS-DeconvNet + DL-ARE (**h**). **i**, Decorrelation axial resolution analysis for image stacks of ER (left), F-actin (middle), and Mito (right) processed by different methods ($n=50$ stacks for each biological structure). Center line, medians; limits, 75% and 25%; whiskers, the larger value between the largest data point and the 75th percentiles plus $1.5 \times$ the interquartile range (IQR), and the smaller value between the smallest data point and the 25th percentiles minus $1.5 \times$ the IQR; outliers, data points larger than the upper whisker or smaller than the lower whisker. Scale bar, $3 \mu\text{m}$ (**a-h**).

We have added above two supplementary figures in the revised Supplementary Materials and related discussions and citations (References No. 38-41) in the revised manuscript (highlighted in Page 12, Lines 271-277).

2. Wording: Line 46/47: "..., enabling significant upgradation of existing FM systems and extension of their applicable range." should read "enabling a significant upgrade of existing FM systems and extension of their application range."

Line 53: "and prone" -> "and are prone".

We thank the reviewer for kindly pointing out these. We have revised them.

Reviewer #2:

Novelty & significance:

1. The manuscript mentions existing supervised deep learning models for SR but lacks a clear comparison with the proposed unsupervised approach. How does ZS-DeconvNet's performance compare to state-of-the-art supervised models in terms of resolution enhancement, image quality, and generalization across modalities? Do we need quantification results on the larger microscopy datasets? The authors didn't mention why RL-based deconvolution is the comparison method when assessing the reconstruction data quality.

We thank the reviewer for these comments and suggestions. The proposed ZS-DeconvNet is an unsupervised image SR technique, of which the prerequisite for implementation is the same as the widely adopted deconvolution algorithms, e.g., RL deconvolution and sparse deconvolution. In contrast, existing supervised DLSR models work only when there is enough high-quality training dataset. Therefore, in our manuscript we mainly compared ZS-DeconvNet with other existing unsupervised methods including RL deconvolution for fair and practical assessment.

To answer the reviewer's question about the comparison of supervised and unsupervised methods, we have added a new Supplementary Fig. 11 to compare the unsupervised ZS-DeconvNet with our previously published supervised SR model DFCA (Qiao, C. et al. Nat. Methods 18, 194-202, 2021) in terms of resolution enhancement, image quality, and generalization. In general, given a large amount of high-quality TIRF/TIRF-SIM training dataset, the supervised DFCA model achieves higher output resolution and PSNR (referring to TIRF-SIM images) than ZS-DeconvNet for both microtubule and lysosome images (Supplementary Fig. 11a-e). However, when applied to data acquired by LLSM, the DFCA model pretrained by TIRF images fails to adapt to LLSM data because of the huge input domain shift in the image plane, pixel size, detection NA, etc., resulting in reconstruction hallucinations and degraded resolution (top right corner in Supplementary Fig. 11f). In this case, ZS-DeconvNet can be retrained or finetuned with the noisy LLSM images themselves, yielding better generalization in practical implementation (bottom right corner in Supplementary Fig. 11f). We have added related discussion in the revised manuscript (highlighted in Page 7, Lines 154-157 and Page 18, Lines 431-434) and figure captions of Supplementary Fig. 11 (highlighted in Page 37 of the Supplementary Materials).

For each biological structures, we generally used ≥ 100 (2D) or ≥ 40 (3D) ROIs augmented from over 35 raw images (2D) or 20 raw image volumes (3D) by random cropping, rotation, and flipping in main and supplementary figures. The specific sample amount (n) of each statistic has been labelled in corresponding figure captions. Moreover, we further constituted a larger dataset including 160 low-high SNR 2D image pairs and 76 low-high SNR 3D image volume pairs (before augmentation) of

three biological structures for each, and systematically quantified the $>1.5\times$ resolution enhancement and $>10\times$ reduction in fluorescence intensity by ZS-DeconvNet (Supplementary Fig. 10, see also Response #3).

Supplementary Fig. 11

Supplementary Fig. 11 | Comparison of ZS-DeconvNet with DFCAN in terms of fidelity, resolution, and generalization. **a, b**, Representative microtubule (a) and lysosome (b) images of low (first column) and high SNR (second column) or reconstructed by ZS-DeconvNet (third column) or DFCAN (fourth column). **c, d**, Statistical comparison of ZS-DeconvNet and DFCAN in terms of PSNR (c, $n=120$) and FRC resolution (d, $n=120$). **e**, Representative SR TIRF images reconstructed by DFCAN (trained with TIRF/TIRF-SIM image pairs) and ZS-DeconvNet (trained with only noisy TIRF images). **f**, Representative SR LLSM single slice image reconstructed by DFCAN (the same model used in e, because there is no isotropic GT-SIM data to train a new model) and ZS-DeconvNet (trained with noisy LLSM images themselves). The comparisons shown in e and f indicate that the unsupervised ZS-DeconvNet can be well-generalized to a new data type without additional training dataset, while the conventional supervised SR model such as DFCAN suffers from the performance degradation due to the domain shift problem. Scale bar, $3\ \mu\text{m}$ (a, b, e, f), $1\ \mu\text{m}$ (zoom-in regions of a, b, e), $2\ \mu\text{m}$ (zoom-in regions of f).

Methodology:

2. The manuscript uses a two-stage network but doesn't delve into its uniqueness. From the methodology, the authors proposed simple image reconstruction networks for the tasks as denoising and deconvolution, which are already a proposed models (example: <https://opg.optica.org/prj/fulltext.cfm?uri=prj-9-5-B168&id=450235>) and using multiple

networks such as U-net for denoising and U-net for deconvolution process, or RCAN networks for the deconvolution process is not a unique contribution.

We are sorry for not describing the conceptual advances of ZS-DeconvNet as well as its differences from existing methods well and causing the reviewer's misunderstanding. There are two intrinsic differences between ZS-DeconvNet and the SR-REDSIM model pointed out by the reviewer (Shah, Z. H. et al. Photonics Research 9, B168-B181, 2021): (i) SR-REDSIM is a supervised method that is trained with paired wide-field and SR-SIM images, which can be used only when there is a SIM system to acquire the GT-SIM data for network training. In contrast, ZS-DeconvNet is an unsupervised method with no requirement on training dataset or imaging systems; (ii) SR-REDSIM was exclusively developed for SIM reconstruction, while ZS-DeconvNet exploits the general imaging model and noise distribution (Supplementary Note 1), which is widely applicable for various microscopy modalities including but not limited to SIM.

In this paper, we do not intend to emphasize the innovation of neural network architectures as our previous publications (Qiao, C. et al. Nat. Methods 18, 194-202, 2021; Qiao, C. et al. Nat. Biotechnology 41, 367-377, 2023) do, but proudly introduce the novel zero-shot learning-based unsupervised SR reconstruction framework. To this end, we adopted relatively simple but effective backbone models of U-net and RCAN as many existing works did (Weigert, M. et al. Nat. Methods 15, 1090-1097, 2018; Li, X. et al. Nat. Methods 18, 1395-1400, 2021; Chen, J. et al. Nat. Methods 18, 678-687, 2021), which, of course, can be replaced by more advanced backbone models to pursue better performance as we have discussed in the manuscript (Page 19, Lines 435-440). Herein, we summarize the unique contributions of ZS-DeconvNet methodology as below:

Theoretical contributions:

- We devised an unsupervised deep learning SR scheme with physics-informed deconvolution loss to simultaneously denoise and deconvolve microscopic images. After in-depth study of the distribution and cross-correlation of the noises in recorruped images, we found that paired noise-independent images can be generated with physically pre-determined parameters from a single microscopic image of any imaging modality (Supplementary Note 1a). Moreover, we theoretically proved the convergence of the recorrupion-integrated deconvolution loss based on the linearity of PSF convolution (Supplementary Note 1a, b). All of these theoretical works inspired us to devise the noise-robust unsupervised ZS-DeconvNet, allowing to enhance microscope image resolution with only one input image even at low SNR conditions.
- To extend the applicability of ZS-DeconvNet to further enhance the resolution of super-resolution image at relatively low SNR condition, after in-depth analysis of the noise propagation through the SIM reconstruction process, we theoretically demonstrated that the expectation of the reconstructed noise in SIM images is zero (Supplementary Note 1c). This allowed us to incorporate the recorrupion-based self-supervised deconvolution scheme with SIM reconstruction, and constructed the ZS-DeconvNet-SIM framework. Moreover, we theoretically proved the convergence of the proposed loss function for ZS-DeconvNet-SIM (Supplementary Note 1c, d), which enables unsupervised denoising and deconvolution for SIM images.

Technological contributions:

- Based on the theoretical exploration described above, we constructed the dual-stage ZS-DeconvNet and ZS-DeconvNet-SIM and performed a series of experiments to validate that these

theoretical capabilities are also efficiently realizable in practice. Unlike previous supervised DLSR models relying on a large amount of high-quality training data, the proposed ZS-DeconvNet takes only the noisy input data for its totally unsupervised training. In particular, with appropriate data augmentation (Methods) and the multi-loss-based joint optimization, the ZS-DeconvNet can be well-trained with only one planar image or stack while outperforming the most commonly used RL deconvolution and the state-of-the-art sparse deconvolution (Nat. Biotech., 40, 606-617, 2022), in both qualitative and quantitative evaluation (Supplementary Figs. 6 and 13).

- To make our methods more accessible and convenient to use, we constructed Java-based Fiji plugin for ZS-DeconvNet. Unlike most existing Fiji plugins for deep learning that can only run models trained by python scripts or elsewhere, we integrated both training and inference functionality into our Fiji plugin and built up a comprehensive homepage for ZS-DeconvNet (<https://tristazeng.github.io/ZS-DeconvNet-page/>), allowing users even without deep learning experience to easily utilize the ZS-DeconvNet method in a click manner.

In fact, the most prominent motivation for us is to develop a practical image postprocessing technique that is helpful and freely accessible for biological researchers. Just like the reviewer #1 commented that “the introduction of ZS-DeconvNet is a significant achievement for the general applicability of deep learning-based microscopy data post-processing for denoising and resolution enhancement”, we believe that after the iterative updates under the advices from the reviewers and other users, the ZS-DeconvNet method and its open-source user-friendly software will be a competent tool for daily bioimaging experiments of users worldwide, and hold substantial potential for shedding new light on diverse biological phenomena.

In the revised manuscript, we have summarized the conceptual advances of ZS-DeconvNet in the Discussion section (highlighted in Page 17, Lines 392-402), and made several revisions to clarify the training data acquisition and usage for ZS-DeconvNet to better distinguish it from existing supervised methods (highlighted in Page 7, Lines 144-147 and Page 12, Lines 263-265).

3. Quantification of the reported 1.5x resolution enhancement and 10x lower fluorescence power should be reported on a large dataset?

Appreciate this suggestion. We have added a new Supplementary Fig. 10 to systematically quantify the resolution enhancement and reduction in fluorescence intensity using a large dataset of four different biological structures, which includes 160 low-high SNR image pairs acquired with the TIRF model of our multi-SIM system (Methods) and 76 three-dimensional image volume pairs acquired by our LLSM system. These results quantitatively suggest a 1.5× resolution enhancement and over 10× reduction in fluorescence intensity for ZS-DeconvNet.

Supplementary Fig. 10

Supplementary Fig. 10 | Resolution and fluorescence intensity quantification for dataset and ZS-DeconvNet enhanced images. **a, b**, Representative 2D (**a**) and 3D (**b**, MIP) noisy input images, high SNR GT images, and ZS-DeconvNet enhanced images of ER, mitochondria, lysosomes, and microtubules. Scale bar, 3 μm . **c**, Lateral resolution comparison by FRC analysis of noisy input images, high SNR GT images and ZS-DeconvNet enhanced images ($n=120$). The diffraction limits for excitation wavelength of 488 nm (for ER and MT, gray dashed lines) and 560 nm (for Lyso, orange dashed line) are labelled for reference. **d**, Comparison of fluorescence intensity between low SNR input images for ZS-DeconvNet and high SNR GT data of which the signal level is typically used in routine SR imaging ($n=160$ for 2D WF, $n=76$ for 3D LLSM). The signal intensity is quantified by averaging sCMOS counts of top 1% pixels for each image or stack. Each line represents a low- and high-SNR pair of 2D images or 3D stacks, showing that the fluorescence intensity of high SNR GT data is more than 10-fold higher than that of noisy input data of ZS-DeconvNet.

Usability:

4. Missing a table to show what pretrained model to use for a specific application? See the attached image during the inference time.

Sorry for the confusion. We have added a table (Fiji_pretrained_models_list.xlsx) to show the correspondence between the pre-trained models and network configurations & test data in the google drive repository (<https://drive.google.com/drive/folders/14Fh2IDcFykoNf7JvbXJIARi-FukW2rqC>).

5. I missed a table to show the list of all trained networks for each sample or modality for 2D and 3D samples, including their training parameters (learning rate, epochs, steps, loss, time for each epoch, results of the improvement in the case of denoising and super-resolution, PSNR, SSIM, and FRC values).

Thank you for this suggestion. We have listed the implementation details of ZS-DeconvNet models used for all figures and videos in Supplementary Tables 1 and 2, including the initial learning rate, training patch size, total training iteration, training time, inference time, etc. Taken together the source codes and tutorial homepage, we believe these substantial instructions and details are enough for users to replicate our methods and results.

6. It is not easy for a common user to understand how to use this FIJI application tool and get quick results within due time to verify whether these models are actually helpful or not.

We thank the reviewer for kindly pointing this out. We previously provided a step-by-step instruction as well as a screen recording to show the training and inference workflow of ZS-DeconvNet in the “Fiji: How to do all above with a click” section of our tutorial homepage (<https://tristazeng.github.io/ZS-DeconvNet-page/Tutorial/>). As suggested by the reviewer, we have further added a “5-step quick guide” section in both the website (<https://tristazeng.github.io/ZS-DeconvNet-page/>) and readme file on Github (<https://github.com/TristaZeng/ZS-DeconvNet/blob/main/README.md>) to help common users run our Fiji plugin with minimal steps and have an intuitive sense of ZS-DeconvNet quickly.

Additional comments:

The work described in the manuscript involves the ZS-deconvolution application on microscopy techniques and genome editing methods to study cellular and developmental biology in different model systems, including cell lines, *C. elegans* embryos, and mouse embryos. There are several additional aspects of the paper that may warrant critical scrutiny and further clarification:

7. Justify the choice of specific percentiles for image normalization and provide clarity on how they were determined.

The percentile normalization is a common post-processing method for high-contrast display and fair quantitative comparison, which has been applied in many existing publications (e.g., Weigert, M. et al., Nat. Methods, 2018; Qiao, C. et al., Nat. Methods, 2021). We tested several p_{low} and p_{high} values in Eq. (15) within the suggested ranges of [1, 3] and [99.5, 100] by these papers, and found that the image contrast was not sensitive to the choice of p_{low} (first three columns in Response Fig. 2), while p_{high} values lower than 100 might cause oversaturation effects (fourth column in Response Fig. 2). Therefore, in our manuscript we selected 3 and 100 for p_{low} and p_{high} , respectively, for all methods.

Response Fig. 2

Response Fig. 2 | Representative images processed by sparse deconvolution and ZS-DeconvNet normalized with different p_{low} and p_{high} values.

8. Enhance clarity in describing cell culture, transfection, and staining protocols, including rationale and references for each step.

We appreciate the reviewer for the careful investigation and these suggestions. In the revised manuscript, we have enhanced the clarity of our cell culture, transfection, and staining protocols by incorporating crucial details, including catalogue numbers of essential reagents, rationale for each step, and references supporting our methodologies. For cell culture, we have provided comprehensive information, including catalogue numbers of essential reagents, to facilitate replication of our experimental procedures. Moreover, our staining protocol has been elucidated with clarity, including detailed staining duration and ligand concentration and rationale for each stage of the procedure. These revisions are highlighted in Pages 31-32, Lines 813-837.

9. Provide details on sgRNA/TALEN design, efficiency, and validation methods to ensure specificity and reliability of genome editing.

The TALEN-mediated editing of the SUM159 cell line (55-CLTA-EGFP-sort2) was donated by Kangmin He, which has been previously reported in a publication as duly cited (He, K. et al. Nature 552, 410-414, 2017). Detailed procedures for the construction of this cell line have been described in the aforementioned paper.

For the insertion of mEmerald into the C terminus of SC35 in HeLa cells, sgRNAs were designed targeting the region near the stop codon of SC35 using the CRISPick webserver. The sgRNA with the highest predicted score was selected and ligated into the pX330A1×2 plasmid (referred to as pX330-SC35-gRNA). Transfection of pX330-SC35-gRNA into HeLa cells was followed by validation of its genomic editing capacity using the T7E1 assay. Subsequently, the donor vector p-SC35-doner, containing mEmerald flanked with approximately 1800bp homology arms complementary to the stop

codon of the human genomic SC35 locus, was constructed and transfected into HeLa cells along with pX330-SC35-gRNA. Cells were then sorted for mEmerald-positive cells using FACS, followed by lentiviral infection with H2B-mCherry and single-cell seeding into 96-well plates. Genomic DNA from different single-cell clones was extracted after two weeks and validated by PCR and Western blot analysis. Homozygous SC35 knock-in cells were selected for further study.

Similar procedures were employed for the construction of the CRISPR dual-edited SUM159 cell line (EGFP-Rab11a-A1 Lamp1-Halo-C8), with EGFP and Halo inserted into the N-terminus and C-terminus of Rab11a and Lamp1, respectively.

To validate the specificity and reliability of genome editing, PCR and Western blot analyses were conducted on the generated cell lines. The results of the validation are depicted in the provided Response Fig. 3, confirming the successful generation of the desired edited cell lines.

Response Fig. 3

Response Fig. 3 | PCR (upper panel) and Western blot (lower panel) analysis of cell lines used in this paper. a, 55-CLTA-EGFP-sort2 SUM159 cell line. b, EGFP-Rab11a-A1 Lamp1-Halo-C8 SUM159 cell line. c, SC35-mEmerald KI HeLa cell line.

10. Include specific details on imaging parameters and analysis methods to assess data quality and validity accurately.

We thank the reviewer for this suggestion. We have listed all primary imaging parameters including the imaging modality, cell type, fluorescence label, excitation NA and wavelength, exposure time, acquisition time, illumination intensity, etc., of every live-cell experiment in Supplementary Table 3. Data post-processing and analysis methods have been included in “Data post-processing and SR image evaluation” section of Methods in the revised manuscript.

11. Expand on why specific methods were chosen (e.g., Lipofectamine 3000 vs. other transfection reagents). Justify any non-standard techniques used.

In the case of transfection reagents, while Lipofectamine 2000 is commonly employed, our decision to utilize Lipofectamine 3000 was based on empirical evidence demonstrating its superior transfection efficiency and reduced cell toxicity across multiple cell lines, including SUM159, HeLa, and COS7. This choice was further substantiated by a comparative analysis, supported by a snapshot obtained from the Invitrogen website (see Response Fig. 4), which highlighted the favorable performance of Lipofectamine 3000.

[figure redacted]

Response Fig. 4 | Snapshots for introduction of Lipofectamine 3000 from the Invitrogen website.

12. Discuss potential off-target effects of genome editing tools and strategies for minimizing or evaluating their impact on experimental outcomes.

Notably, within our genome-edited cell lines, the absence of non-specific fluorescence signals is a significant observation. This absence suggests that off-target effects may not be prominent within our experimental system, providing initial reassurance regarding the fidelity of our results.

Furthermore, despite these promising observations, it remains imperative to implement strategies aimed at minimizing potential off-target effects. To this end, we utilized CasOff (<http://www.rgenome.net/cas-offfinder/>) to predict potential off-target sites for the sgRNAs utilized in our study. Encouragingly, analysis revealed that all three sgRNAs utilized for constructing SC35-KI

cells and EGFP-Rab11a-A1 Lamp1-Halo-C8 dual-edited cells exhibited zero predicted off-target sites across the genome.

Collectively, while comprehensive evaluations such as whole-genome sequencing or targeted deep sequencing were not performed to assess genomic alterations, our findings suggest that the potential off-target effects within the cell lines used in our study are minimal.

13. Provide comprehensive information about reagents, materials, and sources used in experiments to facilitate reproducibility and transparency.

In our efforts to promote reproducibility and transparency, we have meticulously documented and provided comprehensive information regarding the reagents, materials, and sources utilized in our experiments. As previously stated in Response #8, detailed information on the reagents and cell lines employed has been made available. This includes precise specifications regarding the sources, catalog numbers, and any pertinent details necessary for accurately replicating our experimental procedures (highlighted in Pages 31-33, Lines 813-870 of the revised manuscript).

14. Include details on statistical methods used for data analysis, including sample size determination, significance thresholds, and appropriate tests for comparisons.

We thank the reviewer for this suggestion. We used one-way ANOVA with the Dunnett's multiple comparisons test in Fig. 2h with a significance threshold of **** $p < 0.0001$, and have included corresponding description in the figure caption. We used box-and-whisker plots or violin plots in other main figures and supplementary figures without statistical tests or null hypothesis testing, and all information about the center lines, limits, and whiskers have been labelled in the corresponding figure captions. Sample size was not predetermined based on statistical calculations. But for every experiment in this study, we generally performed 30~120 replications to ensure reproducibility. The sample size (n) of each experiment is provided in the figure captions in the main text and Supplementary Materials. Also, we have added these claims in the "Reporting Summary" file.

15. Include specifics like microscope settings, objective lenses, resolution, and exposure times. This allows for replication and evaluation of image quality.

Appreciate this suggestion. As is discussed in Response #10, the microscope settings for each live-cell experiments have been listed in Supplementary Table 3, and the system setups including objective lenses, camera, etc., have been described in the section "Optical systems" of Methods.

16. Describe any image processing steps applied (e.g., filtering, background subtraction). Mention software used and relevant parameters.

Appreciate this suggestion. We have described our image processing steps in the "Data post-processing and SR image evaluation" section of Methods, and we also listed the hyperparameters used in RL deconvolution and sparse deconvolution software in Supplementary Table 2.

17. Briefly outline future research directions based on your findings. This demonstrates the broader significance of your work and stimulates further investigation.

We thank the reviewer for this suggestion. We have outlined the potential future extensions for ZS-

DeconvNet in Discussion section (Page 19, lines 435-450 of the revised manuscript).

18. Include the following state-of-the-art methods relevant to the study: Noise2Noise imageJ plugin for image denoising, RL-based deconvolution for the fluorescence microscopy. There are multiple studies shows the effect of small training datasets for the super-resolution using DeepSTORM, application specific super-resolution methods to show the minimum dataset required for the SR training methods.

Appreciate this suggestion. The relevant citations, comparison, and discussions about Noise2Noise, its recent variant DeepCAD (reference No. 33, 63, Figs. 2a, 3f and Supplementary Fig. 12), conventional RL deconvolution and its recent variant RL-based sparse deconvolution (reference No. 4-6, Figs. 1c, 2a-e, 3c-f, 4a-d) have been included in the revised manuscript.

We emphasize that compared with existing state-of-art methods, the proposed ZS-DeconvNet makes advances in several aspects: **First**, unlike Noise2Noise that requires the other independent acquisition of the same scene as the training target (thus usually dubbed as weakly supervised), ZS-DeconvNet is a completely self-supervised/unsupervised method that can be applied without any additional training data, yielding a wider application scenario; **Second**, compared with other very recent self-supervised denoising techniques such as DeepSeMi (Zhang, G. et al. Nat. Methods 20, 1957-1970, 2023) and SRD-Trans (Li, X. et al. Nat. Computational Science 3, 1067-1080, 2023), ZS-DeconvNet gives not only compatible denoised results but also super-resolved outputs with better spatial resolution and contrast (new Supplementary Fig. 17); **Third**, compared with analytical model such as RL deconvolution or more advanced sparse deconvolution (Zhao, W. Nat. Biotechnology 40, 606-617, 2021), ZS-DeconvNet possesses a superior noise suppression capability owing to its recorruption scheme, resulting in a substantial improvement in output fidelity quantified by PSNR (Fig. 1c, d; Fig. 3c, d); **Fourth**, ZS-DeconvNet is a universal unsupervised denoising and SR technique, which is applicable to a wide range of imaging modalities such as TIRF/WF microscope (Figs. 1 and 4e), confocal microscopy (Fig. 4a-d), light-sheet microscope (Fig. 3), and multi-modal SIM (Fig. 5). In contrast, most existing methods only focus on a specific type of data such as DeepSTORM referred by the reviewer, which is only used to localize single molecules in a sequence of diffraction-limited raw STORM images in a supervised manner.

19. Missing for 2PE/3PE fluorescence microscopy imaging to show the proposed method works on the highly scattering tissues?

Thank you for this suggestion. We have added a new supplementary figure that shows ZS-DeconvNet's performance on two-photon images of a scattering brain slice captured using our home-built two-photon microscope (Zhao, Z. et al. Cell 186, 2475-2491.e2422, 2023). Specifically, a 3D ZS-DeconvNet model was trained with one noisy image stack and then applied to itself for testing and display. By comparing the raw, denoised, and deconvolved images shown in Supplementary Fig. 25a, b, we found that the first stage of ZS-DeconvNet effectively removed the photon noise while the second stage further enhanced the image resolution. Consequently, two adjacent dendritic spines can be distinguished in the deconvolved image output by ZS-DeconvNet, which were blurry in the raw noisy two-photon image (Supplementary Fig. 25c). These results demonstrate the applicability of ZS-DeconvNet on two-photon microscopy when imaging scattering samples. Related content has been added into the main text (highlighted on Page 18, Lines 408-409 of the revised manuscript) and the Supplementary Materials (Supplementary Fig. 25).

Supplementary Fig. 25

Supplementary Fig. 25 | Characterization of ZS-DeconvNet on two-photon microscopy (TPM). **a**, Representative noisy TPM image (left), denoised (middle) and deconvolved image (right) by ZS-DeconvNet. The ZS-DeconvNet model was trained with TPM image stack itself. **b**, Power spectrum coverages of the images shown in **a**. **c**, Intensity profiles along the lines indicated by the arrowheads in the two magnified regions of noisy TPM image (gray) and ZS-Deconvolved image (red) in **a**. Scale bar, 10 μm (**a**), 2 μm (zoom-in regions of **a**).

20. Missing the PSF generation method for each modality and how to handle this PSF phenomenon during the inference stage?

Sorry for this involuntary omission, and appreciate for pointing this out. We provide two methods to generate simulated PSFs: (i) Using the Matlab script we uploaded on GitHub for PSF generation (<https://github.com/TristaZeng/ZS-DeconvNet>); (ii) Using the PSF Generator Fiji plugin licensed by EPFL (<https://bigwww.epfl.ch/algorithms/psfgenerator/>). Of note, in ZS-DeconvNet framework, the PSF, either the real one or simulated one, is only used in the training procedure to calculate the deconvolution loss, and different ZS-DeconvNet models should be trained for different imaging modalities for best performance. After well-trained, the ZS-DeconvNet models can be applied without PSFs during the inference stage.

Moreover, we have added a new Supplementary Fig. 28 to present the consequences of using mismatched PSFs (Supplementary Fig. 28c) as well as other situations where potential hallucinations may appear (Supplementary Fig. 28a, b). The results shown below indicate that ZS-DeconvNet models should be trained using matched PSFs with its input images, otherwise the improperly training with mismatched PSFs might result in lack of resolution improvement or periodically artifacts.

In the revised manuscript and Supplementary Materials, we have supplemented details of PSF generation in “Implementation details of ZS-DeconvNet” section of Methods (highlighted in Page 29, Lines 737-741) and Supplementary Note 4a. We also updated our tutorial homepage to give more instructions about PSF selection and generation.

Supplementary Fig. 28

Supplementary Fig. 28 | Potential hallucinations generated by ZS-DeconvNet. **a**, Representative noisy WF image (first column), ZS-DeconvNet enhanced image (second column), resolution-scaled error (RSE) map (third column) calculated between the noisy WF image and ZS-DeconvNet image, and clear WF image for reference (fourth column). The yellow arrowheads in the magnified images point out an area where ZS-DeconvNet over-weakened the microtubule structures. From our experience, this kind of hallucinations usually happen in regions where the fluorescence intensity in the raw input image is too low, and could be identified to some extent by quality-check tools such as SQUIRREL analysis. **b**, F-actin images (MIP) acquired by LLSM (first column), LLS-SIM (fourth column) and reconstructed by ZS-DeconvNet trained with unmatched TIRF images (second column) and noisy data itself (third column). These results indicate that there might be noticeable performance degradation when applying a trained ZS-DeconvNet model to data of different imaging modalities, e.g., with different pixel sizes, theoretical resolution, background, etc. **c**, Noisy WF images of microtubules (first column) and super-resolved images generated with ZS-DeconvNet models trained with over-narrowed PSF (second column), matched PSF (third column), and over-large PSF (fourth column). These images demonstrate that ZS-DeconvNet models should be trained using matched PSFs with the input images, otherwise the improper training with unmatched PSFs might result in lack of resolution improvement or ringing artifacts. Scale bar, 2 μm (a), 1 μm (zoom-in regions of a), 4 μm (b, c), 1.5 μm (zoom-in regions of b and c).

21. Missing information about the disrupting the noisy images? What is the need, what type of noise is added, what is the distribution of noise added, and what are the PSNR values?

We have detailedly discussed the theoretical basis of image recorrption, noise models adopted, and

recorruption-related parameters in Supplementary Notes 1 and 4. Briefly, we adopted a mixed Poisson-Gaussian noise model for sCMOS camera and approximated it with a Gaussian distribution $\mathbf{g} \sim \mathcal{N}(0, \sigma^2 \mathbf{I})$ during image recorruption procedure, where $\sigma^2 = \beta_1 \mathbf{x} + \beta_2$ with β_1 , β_2 , and \mathbf{x} representing the Poissonian factor, Gaussian factor, and photoelectron count map, respectively. We have validated this assumption from a theoretical perspective (Supplementary Note 1a and Supplementary Fig. SN1) and demonstrated the effectiveness of the image recorruption scheme by conducting numerous experimental and simulation evaluations quantified by PSNR (Figs. 1, 2, and Supplementary Figs. 1, 3-10).

22. Overall, this proposed method works only you have a SR microscopy setup which is an expensive and not a common optical fluorescence bench setup for in vivo imaging.

We are sorry for not describing raw data acquisition and implementation of ZS-DeconvNet well and causing the reviewer's misunderstanding. As is illustrated throughout the manuscript, ZS-DeconvNet is a universal *unsupervised* microscopic image processing method which is compatible with a wide range of imaging modalities from scanning-based microscopy, e.g., confocal microscopy (Fig. 4a-d) and two-photon microscopy (Supplementary Fig. 25), to wide-field detection-based microscopy, e.g., TIRF (Figs. 1, 2), 3D WF microscopy (Fig. 4e-h), LLSM (Fig. 3), and multimodal SIM (Fig. 5). Actually, it is one of the most significant advances of ZS-DeconvNet that it can generate SR images without SR microscopy setups, because both the training and application of ZS-DeconvNet *do not require* either SR optics or SR data as ground truths.

To avoid further misunderstanding of other potential readers, we have added more detailed descriptions about raw data acquisition and training data usage (highlighted in Page 7, Lines 144-147 and Page 12, Lines 263-265) and added the descriptions of the TIRF mode and 3D WF mode of the Multi-SIM system in the Methods section of the revised manuscript (highlighted in Page 24, Lines 600-603).

Comments in the attached manuscript file:

23. L33: Missing for 2PE and light-sheet microscopy, ..., For highly scattering tissue is missing.

We thank the reviewer for this suggestion. ZS-DeconvNet experiments on light-sheet microscopy has been demonstrated in Fig. 3 of the previous manuscript. As suggested by the reviewer, we added a new Supplementary Fig. 25 to validate the applicability of ZS-DeconvNet on two-photon microscopy and scattering brain slice sample (see also response #19).

24. L37: PSF generalization for each modality.

As we have explained above (Response #20), the ZS-DeconvNet should be trained with experimental or simulated PSFs that are corresponding to the specific imaging modality and configuration, which is similar to conventional deconvolution algorithms. After training, ZS-DeconvNet should be applied to the same type of data as training for best denoising and SR performance.

25. L142: what is the training dataset used? how it works for different samples? what happens if the setup doesn't have SR capability?

Sorry for causing this confusion. In the experiments of Fig. 1, we used the low-SNR diffraction-limited images acquired by the TIRF mode of our multi-SIM as the training dataset, and regarded high-SNR

WF counterparts as the reference to calculate PSNR metrics. Actually, we didn't employ the SR capability of the Multi-SIM system to acquire the dataset, but just used the TIRF mode of it. Therefore, the training and inference of ZS-DeconvNet do not rely on any SR capability of imaging systems, just as we clarified in Response #22.

In the revised manuscript, we have specified the dataset usage (highlighted in Page 7, Lines 144-147), and added more details of the TIRF mode and 3D WF mode of the Multi-SIM system in the Methods section (highlighted in Page 24, Lines 600-603).

26. Fig 3A and 3B shows the network as 3D U-net only but not RCAN network

We are sorry for the too similar schematic of U-net and RCAN used in Figs. 1 and 3. We have updated the schematic of RCAN in Fig. 3 and Supplementary Fig. 20 to make it distinguishable enough from U-net in the revised manuscript.

Response Fig. 5 | Schematics of U-net and RCAN in the previous and revised manuscript.

27. L256: number of slices in 3D, z depth resolution, input PSNR, why even slices gave for test and odd only applied denoising?

We thank the reviewer for these comments. All image size information has been listed in Supplementary Table 2, and we added the input PSNR statistics in Fig. 3d of the revised manuscript. Moreover, we have added a new Supplementary Fig. 17 to quantitatively assess the axial resolution improvement brought about by ZS-DeconvNet and compared it with two recently published deep-learning methods for axial resolution enhancement, DL-ARE (Li, X. et al. Nat. Biotechnology, 2023) and Self-Net (Ning, K. et al. Light: S & A, 12, 204, 2023).

Fig. 3d | Statistical comparisons of RL deconvolution, sparse deconvolution and ZS-DeconvNet in terms of PSNR and resolution on different specimens (n=40). The resolution was measured by Fourier ring correlation analysis with F-actin image stacks. Center line, medians; limits, 75% and 25%; whiskers, maximum and minimum.

The training scheme of 3D ZS-DeconvNet integrates our previously proposed spatially interleaved self-supervised (SiS) learning scheme (Qiao, C. Nat. Biotechnology 41, 367-377, 2023) with the self-supervised inverse problem solver. Briefly, each noisy image stack is divided into odd slices and even slices, which are then used as input and targets, respectively. However, to amend the expectation gap between odd and even slices, we introduce the gap amending regularization (GAR, detailed in

Supplementary Note 3e of the NBT paper) into the loss function design (detailed in Supplementary Note 1b). The “denoised stack” boxed in red lines in Fig. 3a is generated without gradient for GAR calculation, thereby it does not need to be propagated into the deconvolution stage.

By reason of the foregoing, to reply the reviewer’s question, the noisy even slices are not given for testing but used as the targets to calculate training loss, while the noisy odd slices go through the whole dual-stage network, generating one group of denoised odd slices and one group of deconvolved odd slices, which are then used to calculate denoising loss and deconvolution loss, respectively (Fig. 3a). In practice, the characters of odd and even slices could be switched following the same principle.

In the revised manuscript, we added more details about the workflow of 3D ZS-DeconvNet as well as the gap amending regularization in “Implementation details of ZS-DeconvNet” section of Methods (highlighted in Pages 28-29, Lines 718-726 of the revised manuscript) and Supplementary Note 1b.

Supplementary Fig. 17

Supplementary Fig. 17 | Comparison of ZS-DeconvNet and other methods in axial resolution improvement. **a**, Representative noisy LLSM images (max intensity projection, MIP) of ER (left), F-actin (middle), and mitochondria (right). **b-h**, XZ-slices, of which the Y-positions are indicated by yellow dashed lines in **a**, from corresponding image stacks without processing (**b**) and processed by Self-Net (**c**), DL-ARE (**d**), sparse deconvolution (**e**), ZS-DeconvNet (**f**), ZS-DeconvNet + Self-Net (**g**), and ZS-DeconvNet + DL-ARE (**h**). **i**, Decorrelation axial resolution analysis for image stacks of ER (left), F-actin (middle), and Mito (right) processed by different methods (n=50 stacks for each biological structure). Center line, medians; limits, 75% and 25%; whiskers, the larger value between the largest data point and the 75th percentiles plus 1.5× the interquartile range (IQR), and the smaller value between the smallest data point and the 25th percentiles minus 1.5× the IQR; outliers, data points larger than the upper whisker or smaller than the lower whisker. Scale bar, 3 μm (a-h).

REVIEWERS' COMMENTS

Reviewer #2 (Remarks to the Author):

thanks for answering all of my questions in the revised manuscript.